# Landmark-based spatial navigation across the human lifespan

**Marcia Bécu[1,2,3]\*, Denis Sheynikhovich[1], Stephen Ramanoël[1], Guillaume Tatur[1], Anthony Ozier-Lafontaine[1], Colas N Authié[4], José-Alain Sahel[1,5,6,7], Angelo Arleo[1]\***

[1]Sorbonne Université, INSERM, CNRS, Institut de la Vision, Paris, France; [2]Kavli Institute for Systems Neuroscience, Centre for Neural Computation, The Egil and Pauline Braathen and Fred Kavli Centre for Cortical Microcircuits, NTNU, Trondheim, Norway; [3]Max Planck Institute for Human Cognitive and Brain Sciences, Leipzig, Germany; [4]Institut de la Vision, Streetlab, Paris, France; [5]Department of Ophthalmology, The University of Pittsburgh School of Medicine, Pittsburgh, United States; [6]CHNO des Quinze-Vingts, INSERM-DGOS CIC, Paris, France; [7]Department of Ophthalmology, Fondation Ophtalmologique Rothschild, Paris, France

**\*For correspondence:**
marcia.becu@gmail.com (MB);
angelo.arleo@inserm.fr (AA)

**Competing interest:** The authors declare that no competing interests exist.

**Abstract** Human spatial cognition has been mainly characterized in terms of egocentric (body-centered) and allocentric (world-centered) wayfinding behavior. It was hypothesized that allocentric spatial coding, as a special high-level cognitive ability, develops later and deteriorates earlier than the egocentric one throughout lifetime. We challenged this hypothesis by testing the use of landmarks versus geometric cues in a cohort of 96 deeply phenotyped participants, who physically navigated an equiangular Y maze, surrounded by landmarks or an anisotropic one. The results show that an apparent allocentric deficit in children and aged navigators is caused specifically by difficulties in using landmarks for navigation while introducing a geometric polarization of space made these participants as efficient allocentric navigators as young adults. This finding suggests that allocentric behavior relies on two dissociable sensory processing systems that are differentially affected by human aging. Whereas landmark processing follows an inverted-U dependence on age, spatial geometry processing is conserved, highlighting its potential in improving navigation performance across the lifespan.

## Editor's evaluation

The findings in the article show that when provided with geometric cues, rather than landmark cues, older adults and children no longer show selective difficulty with learning spatial layouts allocentrically. This important and compelling finding challenges decades of work suggesting that older adults have a selective allocentric navigation deficit. Instead, the findings suggest that older adults may have perceptual issues related to processing and integrating landmarks.

## Introduction

Human navigation strategies and the underlying spatial representations have been the subject of an intense debate (*Burgess, 2006*; *Wang and Spelke, 2002*; *Ekstrom et al., 2017*; *Burgess, 2008*; *Wang et al., 2006*; *Ekstrom et al., 2014*). Wayfinding behavior has extensively been described in terms of two types of navigation strategies, depending on the spatial reference frame in which multisensory representations are encoded. Egocentric strategies rely on spatial codes anchored on the subject's body or the use of visual snapshots of the environment (*Waller and Hodgson, 2006*), whereas allocentric strategies are grounded on representations that are independent from the subject's position

and orientation, akin to a topographic map (*Burgess, 2008*). A large body of experimental work has been devoted to the question of how environmental conditions as well as navigators' individual characteristics influence the strategy preference (*Ekstrom and Isham, 2017*; *Lester et al., 2017*).

The study of age-related differences in human navigation has added a new temporal dimension to this research domain. It has been extensively proposed that a deterioration of neural structures underlying spatial coding during aging would lead to decreased allocentric navigation capabilities in older participants (*Colombo et al., 2017*; *Wiener et al., 2013*; *Moffat, 2009*; *Lithfous et al., 2013*; *Raz et al., 2004*; *Lister and Barnes, 2009*; *Davis and Weisbeck, 2015*; *Driscoll et al., 2005*; *Rodgers et al., 2012*; *Ruggiero et al., 2016*; *Harris et al., 2012*; *Mahmood et al., 2009*). However, the majority of experimental paradigms used to assess allocentric navigation in humans were based on the capacity of subjects to use landmarks (*Wiener et al., 2013*; *Davis and Weisbeck, 2015*; *Driscoll et al., 2005*; *Rodgers et al., 2012*; *Bohbot et al., 2012*), while making geometric cues uninformative about the goal location or preventing subjects from using them.

Here, we postulate that lifetime changes in spatial cognition can be understood in terms of modulation of spatial cue processing capabilities. Hence, an alternative explanation consistent with the literature is that the widely accepted hypothesis of age-related allocentric deficit may in fact reflect landmark processing differences. This view on the spatial navigation capabilities as a function of age leads to a strong prediction: making the geometric layout of an experimental space informative (about the subject and the goal locations in space) should attenuate the egocentric bias in aged navigators and restore the putative allocentric impairment. Besides, since a preference for geometry and a bias toward egocentric strategies were both shown in aging (*Bécu et al., 2020*) and human development as well (*Bohbot et al., 2012*; *van der Ham et al., 2020*; *Newcombe, 2019*; *Nardini et al., 2006*; *Bullens et al., 2010*), polarization of geometry could also improve allocentric spatial behavior in children.

We sought to test these hypotheses by employing a Y-maze experimental paradigm, traditionally used to dissociate egocentric and allocentric navigation in rodents (*Lenck-Santini et al., 2001*; *Rinaldi et al., 2020*) and humans (*Rodgers et al., 2012*). We first comparatively assessed spatial orientation and navigation performances of children, young, and older adults physically moving in an immersive virtual reality environment. Natural, active body and head motion during spatial behavior avoided limitations of joystick-operated paradigms, which prevent subjects, in particular older ones, from integrating visual, proprioceptive, and vestibular cues for reorientation and navigation (*Mahmood et al., 2009*; *Adamo et al., 2012*; *Harris and Wolbers, 2012*). Second, we tested participants in a real-world replica of the Y-maze environment, allowing natural visual inputs to support navigation. Here, we report experimental evidence in support of above predictions by showing that anisotropic geometry eliminates differences between young adults, children, and older subjects in terms of allocentric navigation capabilities. We show that age-dependent differences in wayfinding behavior can then be ascribed to differences in coding landmarks versus geometrical spatial cues. Our findings suggest that the bias toward egocentric navigation in both children and older navigators is conditioned by the spatial cues present in the environment, questioning the traditional view of a specific deficit for allocentric strategies per se in development and aging. Overall, these results highlight the need to revisit the classical allocentric–egocentric dichotomy to encompass the role of two dissociable systems (based on landmark or geometry) in governing spatial cognition across the lifespan.

## Results

We tested 96 participants (29 children, μ = 10, std = 0.49; 22 young adults, μ = 28, std = 4.28; 28 healthy older adults, μ = 73, std = 3.90) in a Y-maze navigation paradigm adapted to study the relative influence of landmark and geometric cues on spatial navigation (79 participants were tested in immersive virtual conditions, while 17 were tested in the real-world replica, see *Supplementary file 2* and *Supplementary file 3* for participant details). The participants were randomly assigned to two groups, and each group was tested in one of two different versions of the Y-maze task. In the classical landmark condition, an equiangular maze was surrounded by three distal landmarks with respect to which any position in the maze could be unambiguously defined (*Figure 1a*). In the novel geometry condition, there were no landmarks and the angle between two of the three arms was set to 50°, making any location in the maze uniquely determined by its anisotropic geometric layout (*Figure 1b*). The experimental protocol was identical in both conditions and consisted of two phases (see 'Materials

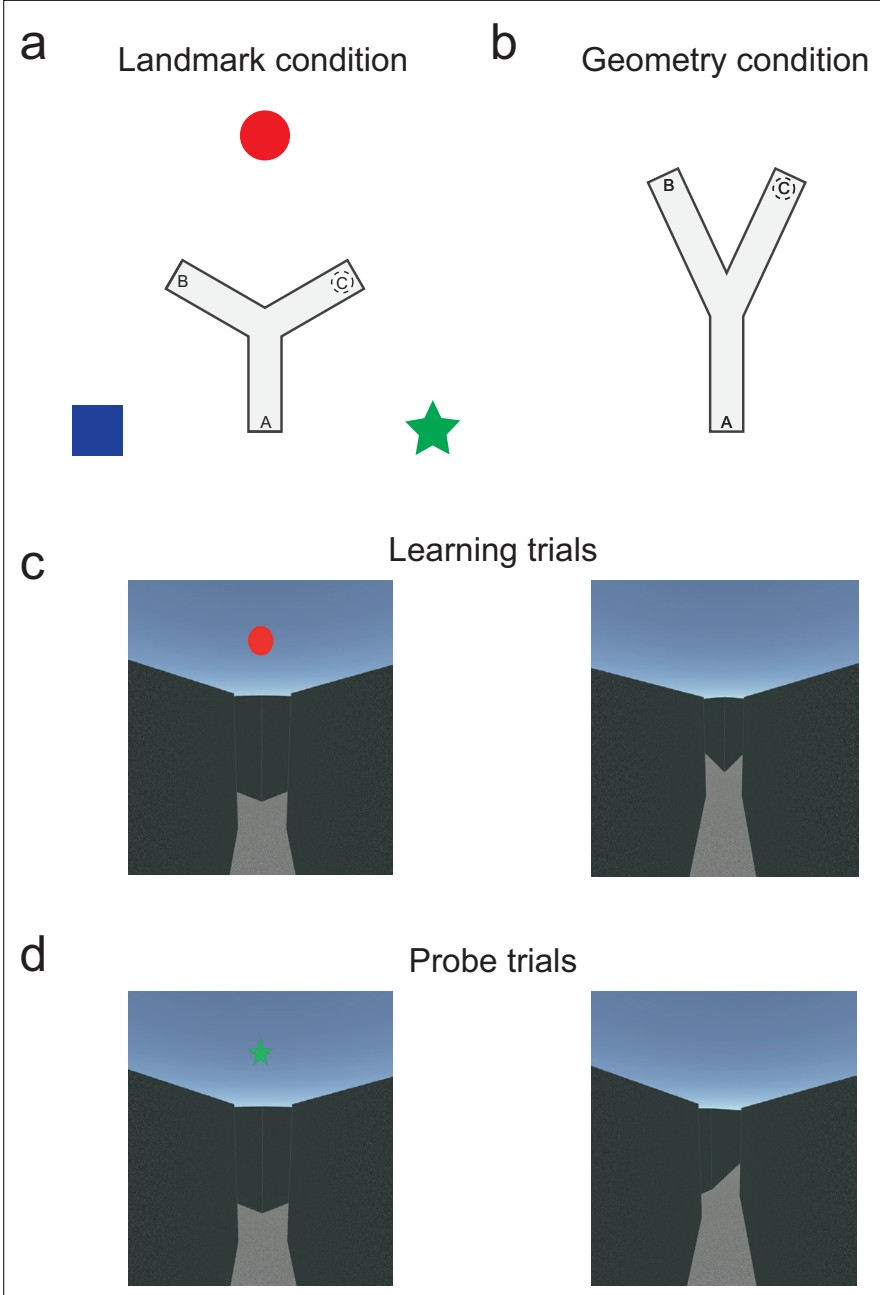

**Figure 1.** Immersive Y-maze tasks to assess the relative influence of landmark and geometric spatial cues as a function of age. (**a**) Top view of the Y-maze during the classical *landmark condition* (i.e., equiangular Y-maze; arm separation: 120°/120°/120°). Three distinct, distal landmarks (blue square, red circle, green star) cued the environment. (**b**) Top view of the Y-maze during the novel *geometry condition* (i.e., anisotropic geometric layout with no landmarks; arm separation: 50°/155°/155°). As depicted in the figure, the corridors in this condition were 54% longer than in the landmark condition to avoid the participants to see the end of the corridors from the starting locations. (**c**) Example of first-person perspective from the departure location during learning trials (i.e., position A in the maze) in the landmark and geometry condition (left and right, respectively). (**d**) Example of first-person perspective from the departure location during probe trials (i.e., position B in the maze) in the landmark and geometry condition (left and right, respectively). See *Figure 1—figure supplement 1* for details on the real-world replica.

The online version of this article includes the following figure supplement(s) for figure 1:

**Figure supplement 1.** Real-world Y-maze implementation.

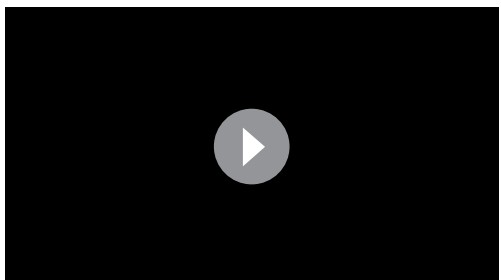

**Video 1.** Disorientation of a participant in immersive virtual conditions.
https://elifesciences.org/articles/81318/figures#video1

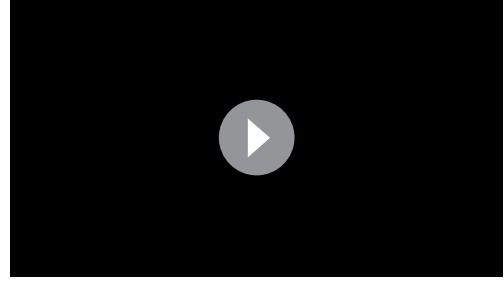

**Video 2.** Examples of learning and probe trials in real-world conditions.
https://elifesciences.org/articles/81318/figures#video2

and methods' for details). All subjects were disoriented with the eyes closed before each trial, and they were placed at the starting location facing the center of the maze (*Video 1*). In the first phase, the subjects learned across multiple trials to navigate from the departure arm (position A, *Figure 1c*) to an invisible target area (position C), reaching of which was notified by sound. Four consecutive successful learning trials triggered the start of the second, testing phase, in which no reward signal was given. The testing phase comprised six trials, in which departure positions were selected among the two non-goal arms: three control trials started from same starting position (arm A) as during learning, while three probe trials started from the arm B (*Figure 1d*). As the subjects were not notified about changes in starting positions, their behavior during the probe trials reflected the strategy they adopted to self-localize and navigate to the goal. If a subject ended up in the arm A during a probe trial, that is, by making the same body turn at the center of the maze as during learning, his/her behavior during that trial was classified as egocentric. If the subject reoriented in space and inferred the position of the target from either the landmark array (landmark condition) or the geometric layout of the maze (geometry condition) and ended up in the goal arm C, the behavior was classified as allocentric. Note that these behavioral classifications were purely descriptive, and that they were chosen to permit the comparison with previous works. They did not mean to capture the nature of processes or representations underlying the associated behavior (*Ekstrom et al., 2014*; *Wang, 2017*).

## Geometric cues enable allocentric navigation in children and older adults

We first assessed strategy preference in the virtual reality settings (n = 79). In the landmark condition, we found a significant bias toward egocentric-like responses in children and older adults compared to young adults (*Figure 2a*; overall Fisher's exact test across the three age groups: p<0.01, Cohen's w = 0.47; children vs. young adults: p<0.01, $\varphi$ = 0.56, odds ratio = 15.80, odds ratio 95% confidence interval (CI) = [1.94:∞], Cohen's w = 0.48; older adults vs. young adults: p<0.01, odds ratio = 14.75, odds ratio 95% CI = [1.88:∞], Cohen's w = 0.45). This result is in agreement with previous reports on allocentric deficits in children and older adults (*Lester et al., 2017*; *Bullens et al., 2010*), while suggesting that the absence of proprioceptive and vestibular cues that characterize desktop-based virtual reality was not the cause for the egocentric bias in previous studies. In the geometric condition of the Y-maze task, there was no significant difference between the three age groups (*Figure 2b*; overall p=0.43; children: p=0.28; older adults: p=0.22). That is, a large majority of children and older adults were able to solve the task allocentrically, similarly to young subjects (see *Figure 2—figure supplement 1b–d* for statistical comparisons of each age group between the two experimental conditions). In order to confirm that the experimental condition differentially affected behavior in the three age groups, we evaluated the statistical interaction between age and condition in a logistic regression model of the data (see 'Materials and methods'). According to this model, the probabilities of making an allocentric response in the landmark condition are 88, 32, and 31% (for young, children, and older participants, respectively), whereas in the geometry condition they are 99, 82, and 82%. The interaction effect is significant (second differences in children vs. young: $\Delta$ = 0.38, p<0.001; children vs. older: $\Delta$ = 0.067, n.s.). These findings suggest that the allocentric deficits in children and older adults

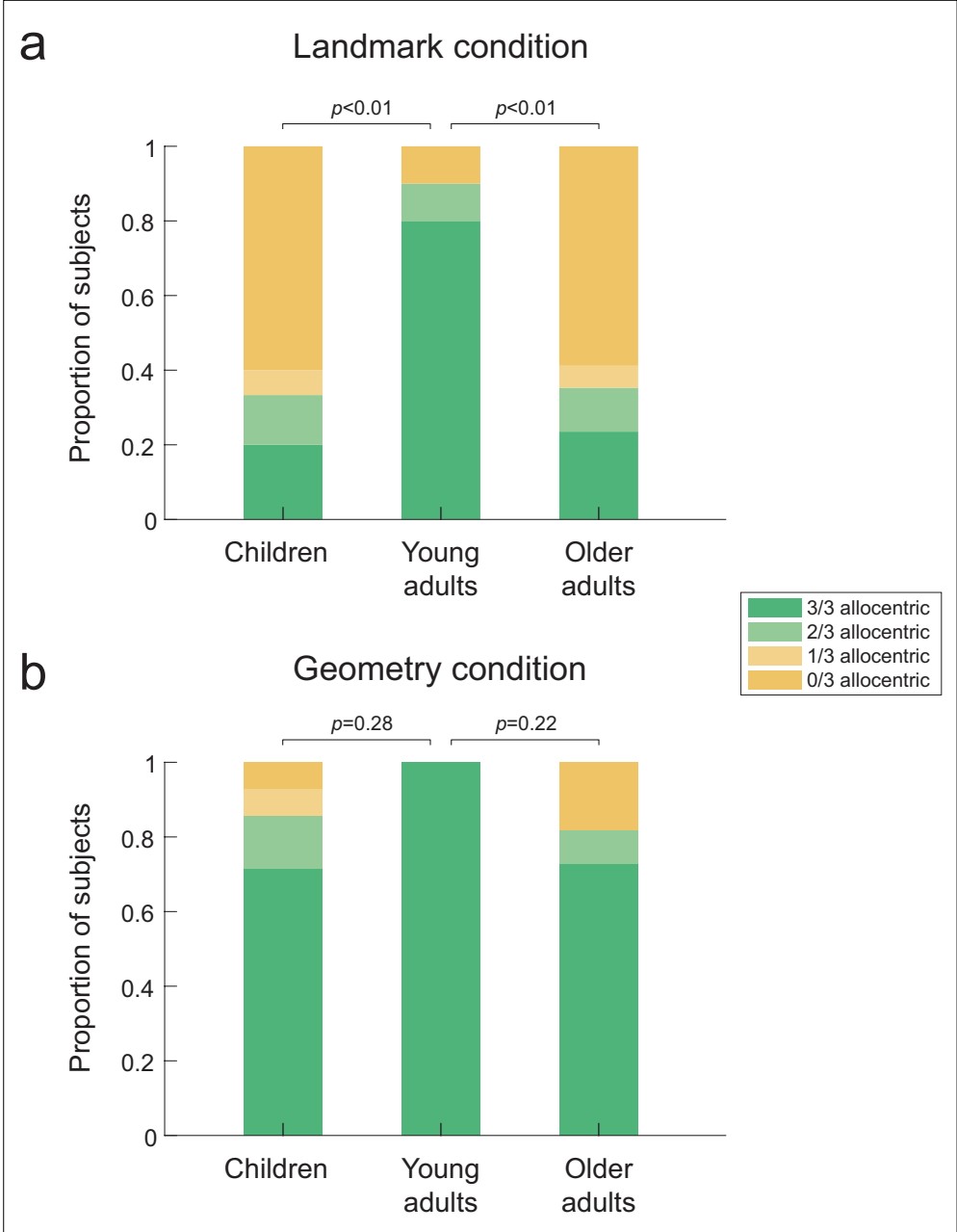

**Figure 2.** Proportion of allocentric behavioral responses during probe trials in the three age groups. Bar plots indicate the proportion of subjects who made either a majority (i.e., 3/3 or 2/3) or a minority (i.e., 1/3 or 0/3) of allocentric choices during the three probe trials. That is, green corresponds to allocentric responses, whereas yellow indicates egocentric behaviors. (**a**) In the landmark condition (n=42), children and older adults failed to solve the Y-maze task since they mostly adopted an egocentric behavior. By contrast, a significant majority of young adults were able to solve the task allocentrically. (**b**) In the geometry condition (n=37), the three age groups behaved similarly, with children and older adults mostly using an allocentric strategy as young adults. p-Values correspond to pairwise comparisons using Fisher's exact test across the corresponding age groups and strategy preferences. The source data for this figure is available in the *Figure 2—source data 1*. This figure corresponds to the strategy preference observed in the virtual reality settings. The same data for the real-world replica can be found in *Figure 2—figure supplement 1a*.

The online version of this article includes the following source data and figure supplement(s) for figure 2:

**Source data 1.** Proportion of allocentric behavioral responses during probe trials across age groups.

**Figure supplement 1.** Proportion of allocentric choices in the three probe trials.

described so far in the literature are linked to a difference in how arrays of landmarks are used to orient and navigate in space. In order to corroborate the results obtained in immersive virtual reality, we reproduced the landmark condition in a real-world replica of the Y-maze in n = 17 participants (see *Video 2*). We found a majority of egocentric responses in older participants navigating the real-world maze (*Figure 2—figure supplement 1a*, p<0.05, odds ratio = 11.11, odds ratio 95% CI = [0.79: ∞], Cohen's w = 0.41), which suggested that the sensory restrictions that characterize immersive head-mounted display (e.g., reduction of the visual field or image quality) were not responsible for the observed egocentric bias. The data from the real-world replica were not further used in the following analyses, unless otherwise specified.

## Landmark-based spatial learning is more difficult for children and older adults

To rule out other sensorimotor or cognitive factors, we comparatively assessed a battery of navigational variables as a function of age, condition and, when possible, interaction of thereof. During learning, subjects could possibly associate the initial view from the starting position with the navigational response (*Figure 1c*). If spatial orientation using landmarks is generally more difficult for children and older adults, this difficulty should be reflected already during learning. To test this hypothesis, we compared the number of trials-to-criterion, traveled distance, escape latency, and navigation speed across the three age groups during learning, in both task conditions. Note that to better interpret the escape latency variable, we also analyzed separately the duration of the orientation period (i.e., the time between the start of the trial and the initiation of locomotion) and the duration of the navigation period (i.e., the time to reach the inferred goal location after locomotion started) for each trial (see 'Materials and methods'). These analyses with age and condition as explanatory factors compared the navigation variables averaged over the first four trials of the learning phase (which were common to all subjects; see also *Figure 3—figure supplement 1* for scatter plots of these data). In the landmark condition, older adults required a higher number of trials than young adults to reach the learning criterion of four consecutive successful trials (*Figure 3a* left; older vs. young adults: U = 287.5, p<0.05, n = 27, $r$ = 0.49, Bayes factor [BF] = 1.97). We observed a similar tendency in children but the statistics were less conclusive (Mann–Whitney U = 226.5, p=0.065, n = 25, $r$ = 0.37, BF = 0.92). In comparison, there was no evidence that children and older adults needed more learning trials to reach to criterion in the geometry condition compared to young adults (*Figure 3a* right; children vs. young adults: U = 213.5, p=0.19, n = 26, $r$ = 0.25, BF = 0.56; older adults vs. young adults: U = 141.5, p=0.55, n = 23, $r$ = 0.12, BF = 0.58). There was a significant main effect of age on navigation variables (traveled distance: $F_{(73,2)}$ = 6.6, p<0.01, BF = 1.53, *Figure 3b*; escape latency: $F_{(73,2)}$ = 8.7, p<0.01, BF = 31, *Figure 3c*; orientation duration: $F_{(73,2)}$ = 3.6, p<0.05, BF = 2.37, *Figure 3d*; navigation duration: $F_{(73,2)}$ = 8.73, p<0.001, BF = 19.88, *Figure 3e*; walking speed: $F_{(73,2)}$ = 3.3, p<0.05, BF = 1.94, *Figure 3f*). The condition strongly influenced the distance traveled by the participants (traveled distance: $F_{(73,1)}$ = 48.6, p<0.0001, BF = 1.1 × 10$^6$) and their walking speed ($F_{(73,1)}$ = 27.9, p<0.0001, BF = 4.7 × 10$^3$), due to the corridors in the geometry condition being slightly longer than in the landmark condition (see 'Materials and methods'). The other navigation variables were minimally or not influenced by condition (navigation duration: $F_{(73,1)}$ = 4.9, p<0.05, BF = 1.21, escape latency: $F_{(73,1)}$ = 1.5, p=0.23, BF = 0.39; orientation duration: $F_{(73,1)}$ = 1.7, p=0.20, BF = 0.47). Finally, evidence in favor of an interaction between age and condition was found for the time-related variables like escape latency ($F_{(73,2)}$ = 3.2, p<0.05, BF = 1.49; *Figure 3c*), navigation duration ($F_{(73,2)}$ = 4.1, p<0.05, BF = 2.56; *Figure 3e*) but not for walking speed ($F_{(73,2)}$ = 2.7, p=0.07, BF = 1.25; *Figure 3f*), traveled distance ($F_{(73,2)}$ = 1.6, p=0.19, BF = 0.51; *Figure 3b*), or orientation duration ($F_{(73,2)}$ = 0.32, p=0.72, BF = 0.22; *Figure 3d*). Further investigation of the interactions showed evidence of a simple effect of age in the landmark condition (escape latency: $F_{(73,2)}$ = 3.1, p=0.052, with Bonferroni correction p=0.10; navigation duration: $F_{(73,2)}$ = 11.3, p<0.0001, with Bonferroni correction p<0.001) but not in the geometry condition (escape latency: $F_{(73,2)}$ = 0.83, p=0.44, navigation duration: $F_{(73,2)}$ = 1.52, p=0.22).

To summarize, in the landmark condition children and older adults adopted direct paths to the goal as young adults, but it nevertheless took them longer, likely reflecting a lower confidence in taking decision when facing the environment composed of landmarks (see below for the analysis of oculomotor behavior supporting this conclusion). Age differences were mitigated in the geometry condition as indicated by the absence of simple effects. In addition, the rapid convergence of

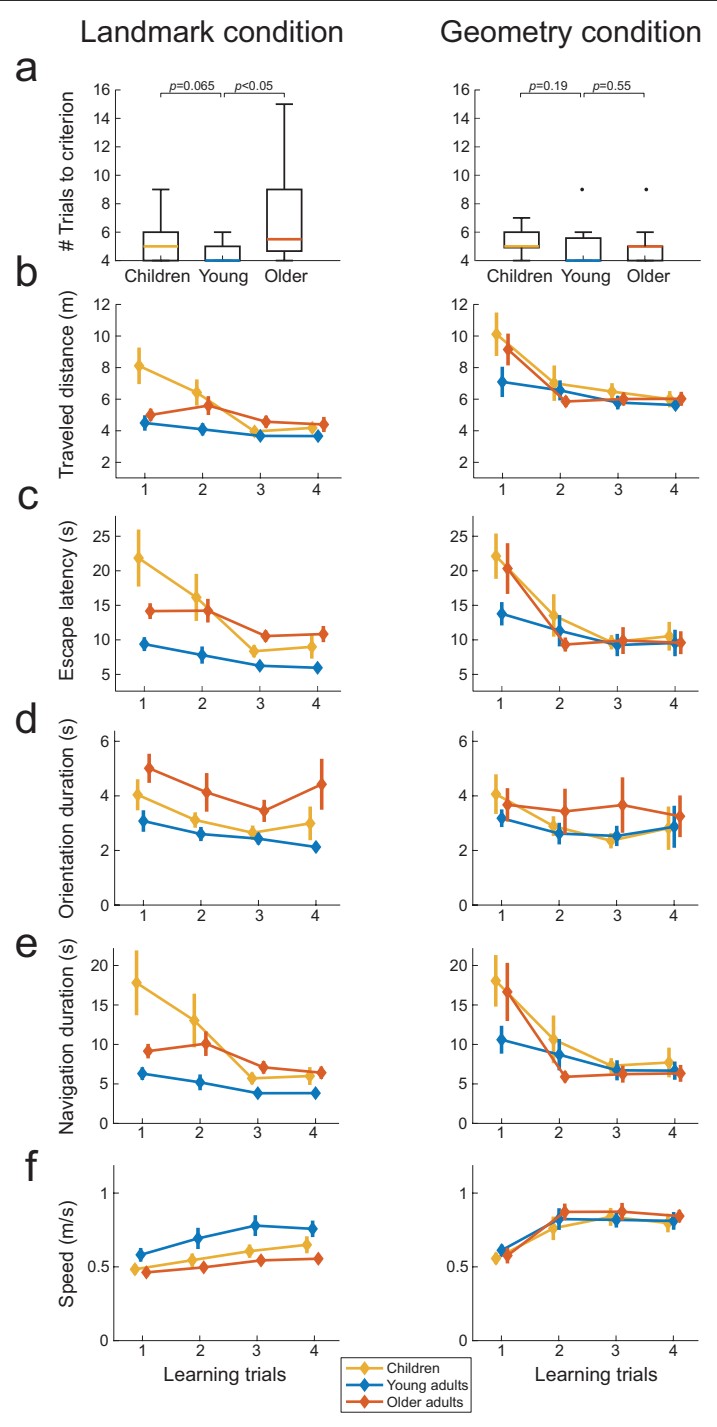

**Figure 3.** Spatial navigation performance during learning trials across the three age groups in the landmark (n=42, left) and geometry (n=37, right) conditions (**a–f**). Colored lines represent median values for the three age groups. Box plots in (**a**) show the median (colored lines), the interquartile range (25th and 75th percentiles, length of the boxes), 1.5× interquartile range (whiskers) and outliers (dots). Error bars represent the standard error of the mean. p-Values in (**a**) shows uncorrected two-samples Mann–Whitney U tests.

The online version of this article includes the following figure supplement(s) for figure 3:

**Figure supplement 1.** Scatter plots of the navigation variables for the first four trials of the learning phase across the three age groups.

learning curves for children and older adults in the presence of geometry (see *Figure 3b, c, e, and f*) suggested that one-trial learning took place in this condition. These results supported the hypothesis that learning to orient using landmarks was more difficult for children and older participants compared to young adults.

## Age-related navigational differences are not due to a faulty attention to landmarks

The combined eye- and head-motion recordings provided a mean to analyze gaze dynamics during reorientation and navigation periods, such as to infer the visual information used by the subjects (*Bécu et al., 2020*). We sought to understand to what extent age-related differences in landmark-based behavior could be associated with specific oculomotor patterns. We only compare here the VR-based gaze dynamics of young against older adults (eye movements were not recorded in children, see 'Materials and methods'). In particular, we investigated whether older subjects' egocentric bias during testing was due to a lack of attention to landmarks during learning. We identified the visual stimuli fixated by each subject by calculating the intersections between the gaze vector and the elements of environment (i.e., landmarks, sky, maze walls, maze floor, etc.; see 'Materials and methods'). In the landmark condition, both young and older participants spent a larger proportion of time at visually exploring the sky region (*Figure 4a*; U = 986, p<0.0001, *r* = 0.82, n = 50, BF = 3311). While looking at the sky region, they focused their visual attention on the landmark that was located directly in front of their departure position (*Figure 4d*). In the geometry condition, conversely, the participants observed mainly the floor region (*Figure 4b*; U = 856, p<0.0001, *r* = 0.74, n = 50, BF = 968), focusing on the lower fork area of the Y-maze (*Figure 4e*). The average time spent on the sky and floor regions in the two conditions was about 20%, with the remaining 80% of the time devoted to fixating maze walls, independently of the condition (*Figure 4c*; U = 718, p=0.57, *r* = 0.08, n = 50, BF = 0.30). Separate analyses for the orientation and navigation periods of the learning trials showed that the sky and the floor were observed mainly during the orientation period, while during navigation the participants looked preferentially at the maze walls (*Figure 4—figure supplement 1*). Importantly, we did not find evidence for an age difference in the time spent gazing at landmarks (*Figure 4a*, U = 119, p=0.30, *r* = 0.27, n = 27, BF = 0.55), suggesting that age-related strategy difference during probe trials did not result from a lack of visual attention during learning.

To further support this conclusion, we analyzed within-group oculomotor signatures. We tested whether there existed differences between the gaze patterns of older subjects that adopted different navigation strategies during probe trials (i.e., allocentric vs. egocentric responses; see 'Materials and methods' and *Supplementary file 1* for the distribution of subjects across age groups and strategy preferences). We found that during learning in the landmark condition, egocentric and allocentric older participants seemingly spent an equivalent proportion of time gazing at the sky region, and in particular at the landmark in front of their starting arm (*Figure 4f*; U = 84, p=0.96, n = 16, BF = 0.44). Similarly, in the geometry condition, we found evidence that egocentric and allocentric older participants gazed at the floor for an equivalent amount of time (*Figure 4g*; U = 13, p=0.90, n = 11, BF = 0.54). Moreover, in both conditions, we did not find evidence for different gaze patterns between allocentric young participants and allocentric or egocentric older participants (*Figure 4f and g*; landmark condition: U = 181, p=0.13, *r* = −0.30, n = 25, BF = 0.67; geometry condition: U = 122, p=0.56, *r* = −0.12, n = 23, BF = 0.43). Note that because there was only one egocentric young subject, we only considered allocentric young participants for these age-based comparisons. These data reinforced the conclusion that the bias toward egocentric navigation responses in the older population was not caused by a lack of visual attention to landmarks during learning. Nevertheless, a majority of them did not use this information during the probe phase.

## Older adults use a view-matching strategy when the landmarks are present

In a subsequent set of analyses, we looked for oculomotor signatures of distinct navigational responses during the probe trials. The rationale was to provide some insights into the dynamical use of spatial cues when the departure arm was different from the one used throughout learning. Gaze dwell-time analyses showed that allocentric navigators spent about 40% of the trial duration exploring the sky region, with no age effect (*Figure 5a*; Mann–Whitney U test, young allocentric vs. older allocentric

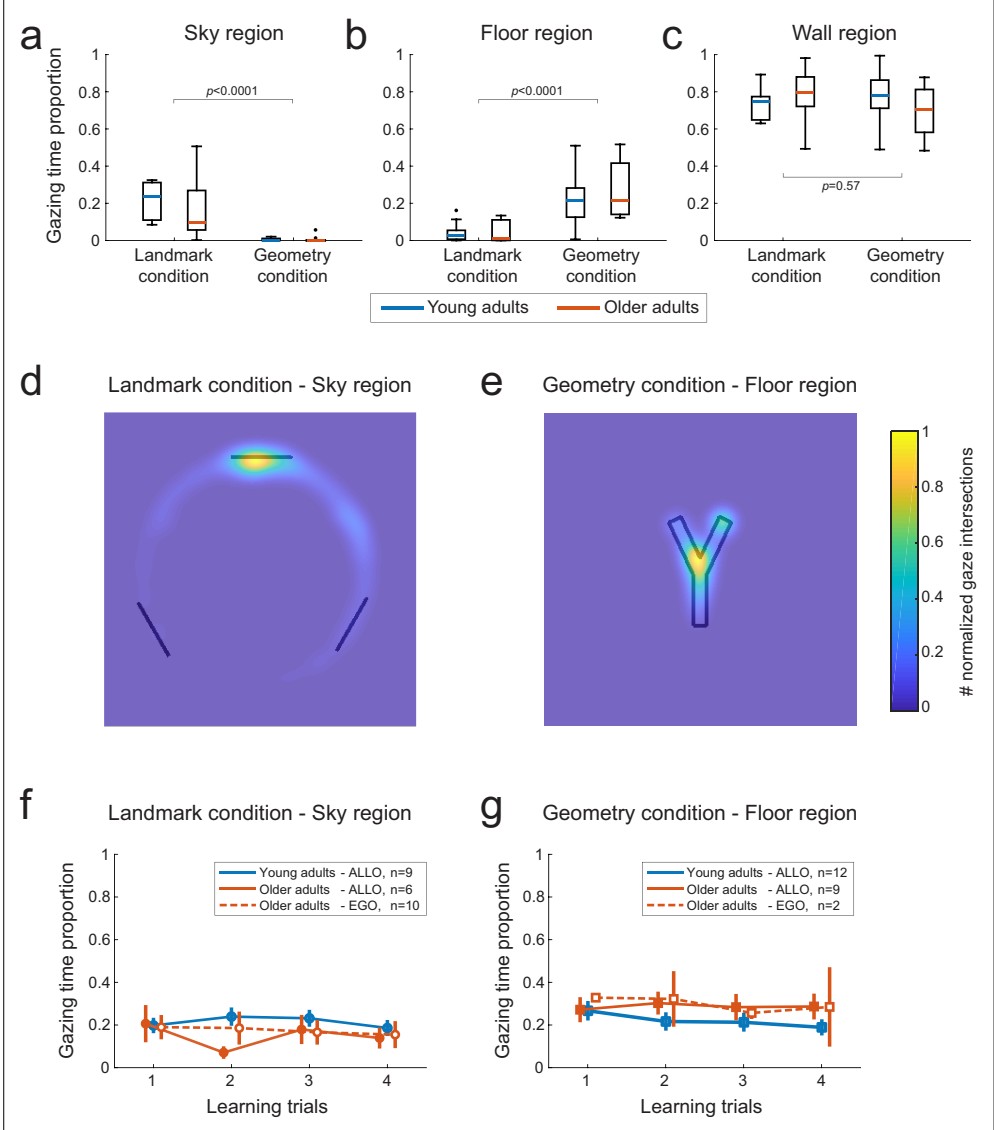

**Figure 4.** Gaze-mediated exploratory behavior during spatial learning. (**a–c**) Gaze dwell-time proportion for sky (**a**), floor (**b**), and wall (**c**) regions of the virtual space as a function of age (young adults n=22, older adults n=28) and experimental condition (landmark n=27, geometry n=23). We found a double dissociation between the time spent at visually exploring sky and floor regions in the landmark and geometry conditions (**a** and **b**, respectively). Neither age nor condition affected the gaze time proportion relative to the walls of the maze (**c**). Data were averaged across the four first trials. Box plots (**a–c**) show the median (colored lines), the interquartile range (25th and 75th percentiles, length of the boxes), 1.5× interquartile range (whiskers) and outliers (dots). (**d**) In the landmark condition, the spatial distribution of the visual focus of attention over the sky region showed that subjects gazed mostly at the landmark facing the departure point. Heatmaps data were pooled across age and the color bar normalization was computed for each group separately. (**e**) In the geometry condition, subjects mostly focused on the fork area of the Y-maze floor. (**fg**) Gaze dwell-time proportion in the sky region of the landmark condition(**d**) and the floor region of the geometry condition (**e**) as a function of learning trials, for young and older allocentric and egocentric subjects. No significant difference existed as either a function of age or strategy preference. Error bars show standard error of the mean.

The online version of this article includes the following figure supplement(s) for figure 4:

**Figure supplement 1.** Oculomotor behavior in the learning phase in young and older groups.

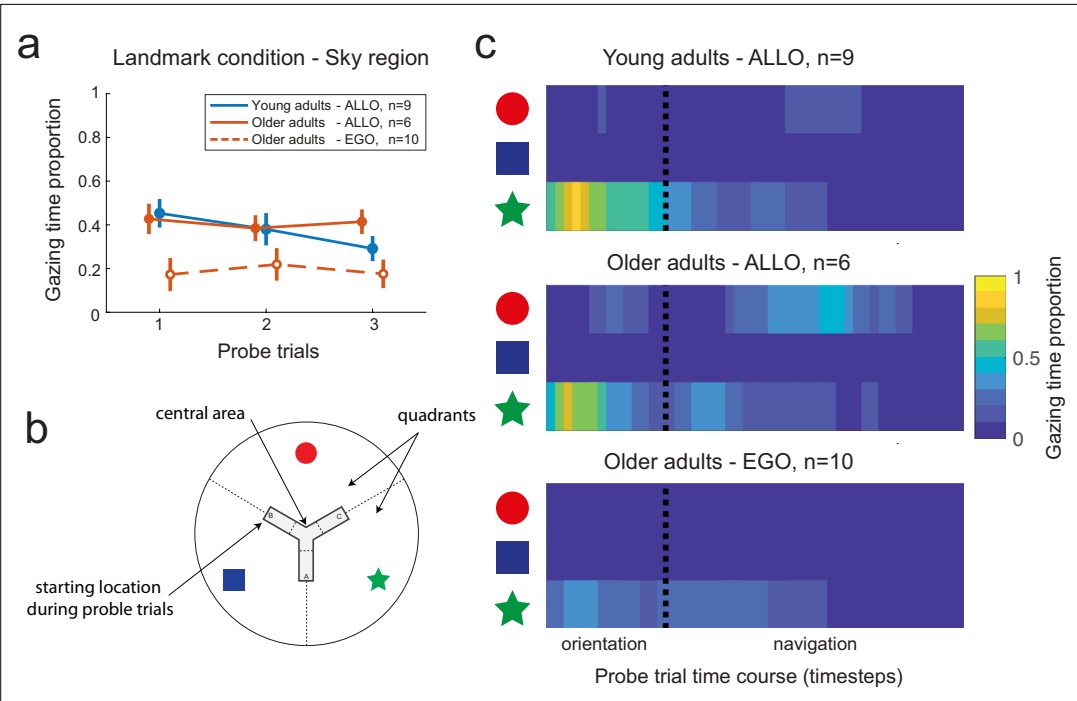

**Figure 5.** Gaze dynamics in the probe trials of the landmark condition. (**a**) Gaze dwell-time proportion relative to the sky region for young and older subjects. Independently from age, allocentric navigators explored significantly more the sky region in the probe trials (~40% of the trial) compared to the learning trials (~20% of the trial, see *Figure 4f* for a comparison). This result did not hold for older egocentric subjects, who spend ~20% of the trial gazing at the sky, irrespective of the learning or probe phases. Error bars show standard error of the mean. (**b**) For analysis purposes, the sky region was separated in landmark-centered sectors, as indicated by dashed lines. (**c**) Evolution of gaze dwell-times throughout the probe trials, including orientation and navigation periods, as a function of landmark sectors, age, and navigation strategy. The star sector corresponds to the landmark directly in front of the departure position in probe trials, while the circle sector corresponds to the landmark directly in front of the starting position in the learning trials. Allocentric young and older subjects focused on the star upon opening the eyes to reorient in space and plan their goal-oriented trajectories. During navigation, allocentric older adults switched their visual focus of attention onto the red landmark when being at the center of the maze. Egocentric older adults looked at the star during orientation as well as while navigating toward the center of the maze.

The online version of this article includes the following figure supplement(s) for figure 5:

**Figure supplement 1.** Time spent gazing at the three landmarks during the first probe trial of the landmark condition (young adults, n=10 and older adults, n=17).

participants: U = 49, p=0.95, n = 15, BF = 0.47). This was about twice the time spent by allocentric participants gazing at the same sky region during learning trials (*Figure 4f*; Wilcoxon signed-rank test, young subjects: W = 44, p<0.01, n = 9, BF = 16; older subjects: W = 21, p<0.05, n = 6, BF = 11). These results were in a stark contrast with those with egocentric subjects, for whom there was no difference in the time spent gazing at the sky area during learning and probe trials (*Figure 4f* and *Figure 5a*; Wilcoxon signed-rank test, older group: W = 32, p=0.69, n = 10, BF = 0.39). These findings suggested that the longer time spent by allocentric subjects at fixating landmarks during probe trials may reflect cognitive processes related to landmark-based spatial orientation. In order to gain more information about these processes, we analyzed the evolution of gaze dwell-times throughout probe trials, during both orientation and navigation periods (by separating the sky area in three sectors corresponding to the circle, square, and start landmarks, *Figure 5b*). We found that gazing at the star (located directly in front of the starting position B) at the very beginning of the orientation period (i.e., upon opening of the eyes) was sufficient for allocentric young subjects to navigate to the goal (*Figure 5c*, top). They did not focus on other landmarks, indicating a good knowledge of the environment in this age group (see also *Figure 5—figure supplement 1a*). Allocentric older navigators had a similar gazing

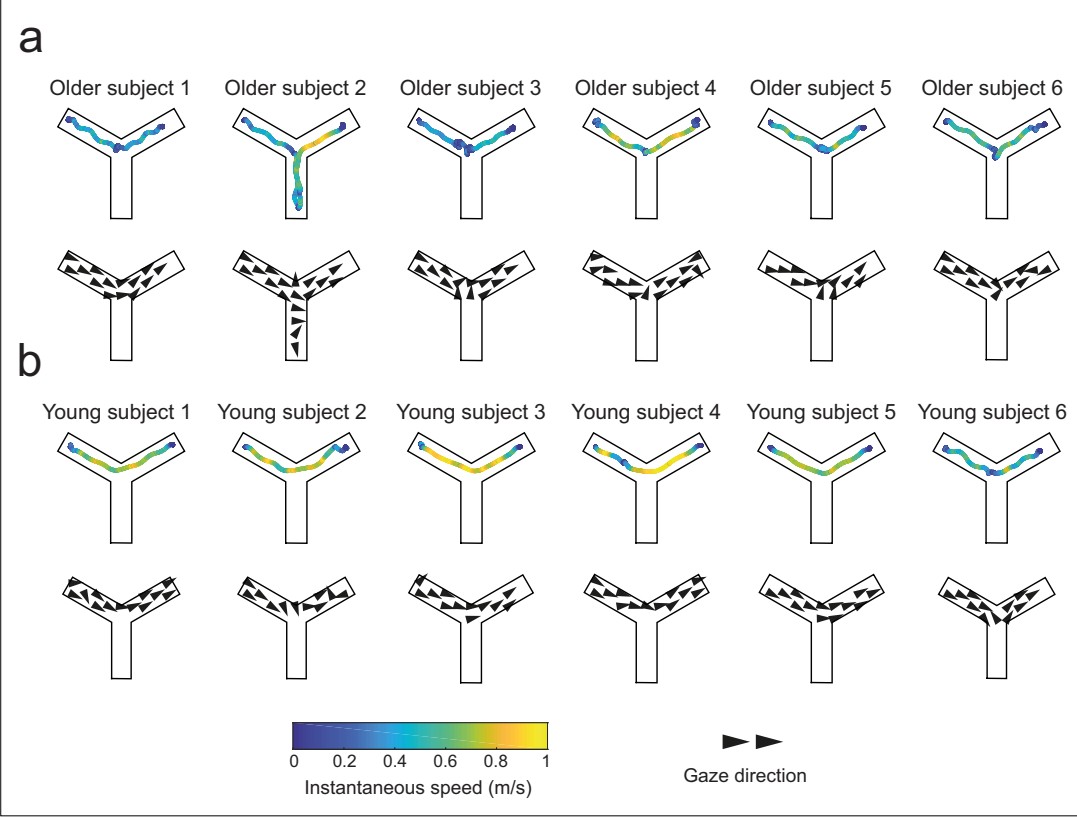

**Figure 6.** Trajectories and gaze vector field representations in the first probe trials of the landmark condition in young and older adults. Qualitative representations of goal-oriented trajectories color-coded with instantaneous speed (top rows), and gaze vectors (bottom rows) of six representative young (**a**) and older (**b**) allocentric navigators. Older adults tended to slow down at the center of the Y-maze where they eventually gazed at the red circle.

behavior during reorientation, but they also looked at the circle during navigation (*Figure 5c*, center and *Figure 5—figure supplement 1b*). Since the circle was directly in front of the departure position during learning, this suggested that allocentric older participants might use a cue-based (i.e., view-matching), rather than map-based, strategy. To verify this hypothesis, we first compared the navigation trajectories of young and older allocentric participants during the first probe trial (*Figure 6a*). We observed that allocentric older navigators tended to decrease their walking speed at the center of the maze and they eventually gazed at the red circle, unlike young adults (*Figure 6a and b* and *Figure 5—figure supplement 1c*). This observation was confirmed by a quantitative analysis showing that when older adults were in the central area of the maze, they spent a longer time gazing at the circle landmark, compared to young ones (U = 45, p=0.054; n = 14, BF = 2), favoring an egocentric view-matching strategy with aging.

## The presence of geometry eliminates age-related differences in navigational performances

We then quantitatively assessed the 'cost' of implementing an allocentric strategy by comparing the navigational variables of young vs. older allocentric participants during probe trials (*Figure 7a–d*). These analyses were carried out in n = 36 allocentric adults, with two-way ANOVAs with Age and Condition as explaining factors or Mann–Whitney rank-sum tests. Interactions were significant (escape latency: $F_{(32,1)}$ = 9.92, p<0.01, BF = 4.5, *Figure 7a*; orientation duration: $F_{(32,1)}$ = 8.36, p<0.001, BF = 4.36, *Figure 7b*; central area: $F_{(32,1)}$ = 13.03, p<0.01, BF = 19, *Figure 7c*), and simple effects with Bonferroni adjustment supported a different impact of age in the landmark and geometry conditions. In the landmark condition, older allocentric adults were significantly slower to initiate walking upon opening the eyes compared to young allocentric subjects (i.e., the orientation duration was longer,

*Figure 7b*; $F_{(32,1)}$ = 20.13, p<0.001), they traveled a longer goal-directed distance (*Figure 7d*; U = 49, p<0.01, n = 15, BF = 2.7), and as a result, they were slower to reach the goal (i.e., larger escape latency, *Figure 7a*; $F_{(32,1)}$ = 30.4, p<0.0001). Coherent with a view-matching strategy in the presence of landmarks, we found that older allocentric subjects spent a longer time (up to 33 s) in the central area of the maze compared to young participants (*Figure 7c*; $F_{(32,1)}$ = 21.13, p<0.001). In the geometry condition, there was no age effect on navigation during the probe trials. Older adults were as quick as young ones in using an allocentric strategy in relation to the geometry, to make their decisions and to reach the goal, which argues in favor of similar processes governing spatial behavior in these two age groups in the geometry condition (*Figure 7a–d* and *Figure 7—figure supplement 1*, escape latency: $F_{(32,1)}$ = 0.92, p=0.68, time in the central area: $F_{(32,1)}$ = 034, p=1, orientation duration: $F_{(32,1)}$ = 0.72, p=0.79, traveled distance: U = 109, p=0.11, n = 21, *r* = −0.35, BF = 0.97).

## Gaze dynamics predicts behavioral strategy during probe trials

Given the clear distinction between egocentric and allocentric subjects in terms of sky vs. floor gaze dwell-times, we sought to assess to what extent eye-movement signatures before the initiation of locomotion could predict future behavioral responses. To do so, we trained a binary classifier (on a single-subject-single-trial basis) with the altitude of the gaze during orientation of the probe trials as the independent predictor variable (see 'Materials and methods'). This choice was based on the observation of more distinct gaze altitude profiles during the orientation period of both experimental conditions (*Figure 8a and b*). We quantified the performance of the classifier by both a 25% hold-out and a leave-one-out validation procedure. We found that gaze altitude patterns during orientation in the landmark condition allowed the future spatial strategy adopted by the subject to be reliably predicted (*Figure 8c*; 25% hold-out: p=0.05, 88 and 61% of participants using an allocentric or egocentric strategy were correctly classified, respectively; leave-one-out: 79% of the subjects were correctly classified, n = 26). Expectedly, the gaze altitude also provided a robust predictor of the experimental condition undertaken by the subject (*Figure 8d*; 25% hold-out: p=0.0001, 97 and 81% of participants were correctly classified in the geometry and landmark condition, respectively; leave-one-out: 88%, n = 49). Supposedly, the lower predictability found in the landmark group, especially in people using an egocentric strategy, reflects the higher variability in gaze altitude observed in this group (see standard errors in *Figure 8a and b*).

## Visual and cognitive correlates of age-related behavioral differences

A subsample of 64 participants (29 young and 35 older adults) enrolled in the study underwent a battery of visual and neurocognitive screenings (see 'Materials and methods' and *Supplementary file 3*). We pooled participants from the VR/real-world experiments and the landmark/geometry conditions to gain statistical power for these visual and cognitive analyses. These tests were not performed in the children group. Among the selected participants, 47 were screened across the complete battery of 19 assessments, while the rest had one or more missing measurements. We exploited these multivariate data to better interpret individual behavioral responses and to avoid potential biases. We first tested whether age and strategy categories corresponded to localized regions of the multivariate feature space. Principal component analysis (PCA) revealed that the participant scores along the first principal axis of the data allowed subjects to be discriminated according to their age (*Figure 9a*; young vs. older: t(45) = 6.32, p<0.001) as well as to the strategy preference in the older population (egocentric vs. allocentric: t(30) = 3.11, p<0.01). The age effect was indeed significant in the large majority (74%) of the screening tests (see *Figure 9—figure supplements 1–2*). We then tested whether the observed strategy differences between experimental conditions (*Figure 2* and *Figure 2—figure supplement 1a*) could possibly be associated to differences in visuo-cognitive characteristics of participants ('Condition effect'), and whether egocentric and allocentric subjects could be distinguished based on their screening test scores ('Strategy effect'). We performed this analysis only in the older adults because there was no egocentric young adult in the geometry condition. We did not find any differences in the test scores according to the task condition (Condition effect, all p>0.0.5), suggesting no visuo-cognitive sampling bias in the data. We found that egocentric older subjects, compared to allocentric ones, had lower scores in the perspective taking test (*Figure 9b*; F(33,1) = 4.34, p<0.05), mental flexibility (B-A difference: F(34,1) = 7.96, p<0.01, B-A ratio: F(34,1) = 9.73, p<0.01; *Figure 8c*), and contrast sensitivity, especially at high frequencies (*Figure 8d*; two circles per degree, 4 cycles per

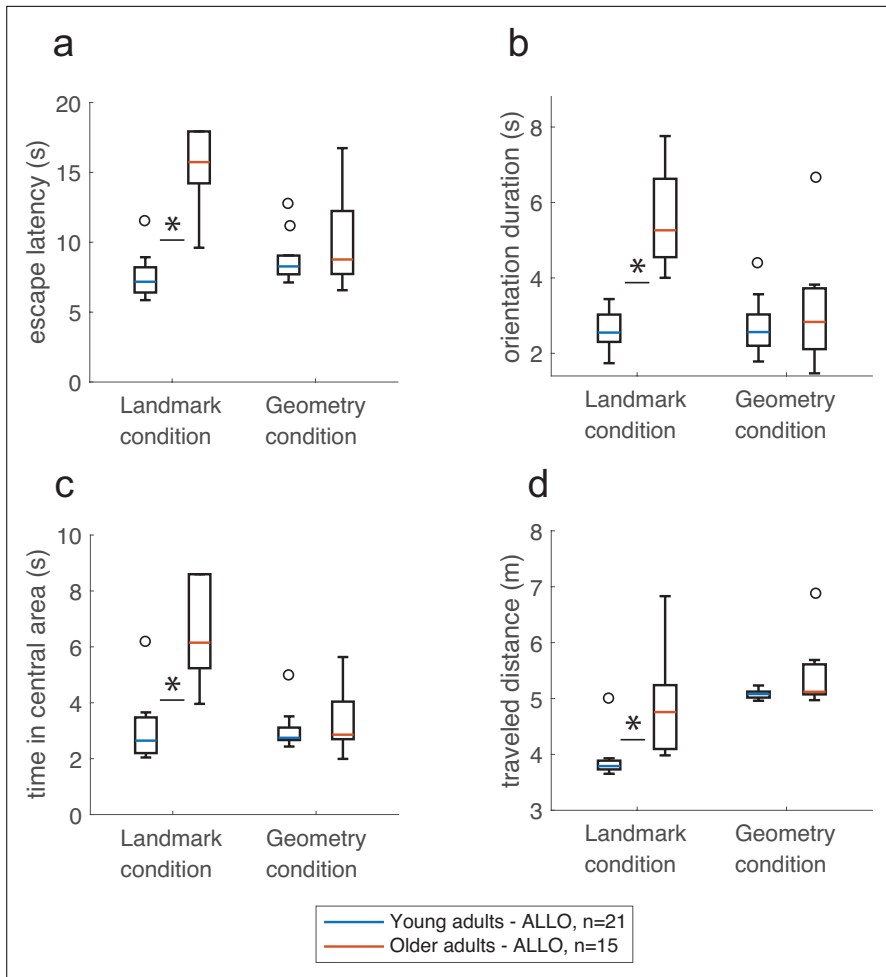

**Figure 7.** The presence of geometric cues eliminated the effect of age on navigation. In the landmark condition, older allocentric adults took longer to reach the goal (**a**), were slower at reorienting in space (**b**), spent significantly more time in the central area of the maze (**c**), and their trajectories to the goal were longer compared to young allocentric adults (**d**). In comparison, there was no age difference in the geometry condition. Box plots in show the median (colored lines), the interquartile range (25th and 75th percentiles, length of the boxes), 1.5× interquartile range (whiskers) and outliers (circles). Stars indicate significant simple effect or Mann-Whitney test. These analysis were carried out in n=36 allocentric adults.

The online version of this article includes the following figure supplement(s) for figure 7:

**Figure supplement 1.** Oculomotor behavior and navigation measures in the probe trials of the geometry condition.

degree [CPD]: U = 194, p=0.088; 8 CPD: U = 187.5, p=0.0042; 16 CPD: U = 184.5, p=0.0029, with Bonferroni $\alpha$ = 0.017, see comparisons for all measures in *Figure 9—figure supplements 1–2*). The main conclusion from these analyses was that the observed deficits in landmark-based spatial coding were likely to be linked to difficulties in flexibly processing and reasoning about landmarks (possibly related to perspective taking, mental flexibility/rotation), as well as to a lower capacity to perceive fine details. In contrast, visual attention, figure memory, and processing speed differences did not seem to play a role.

Finally, we looked for representational errors that could have precluded children and older subjects to use landmarks for orientation. To do so, we tested a subset of participants for their memory of the maze immediately after they completed the landmark condition. We asked 13 children and 7 older adults to (i) recognize the maze shape out of three possibilities (*Figure 9—figure supplement 3a*), (ii) recognize the landmarks within an ensemble with three distractors (*Figure 9—figure supplement*

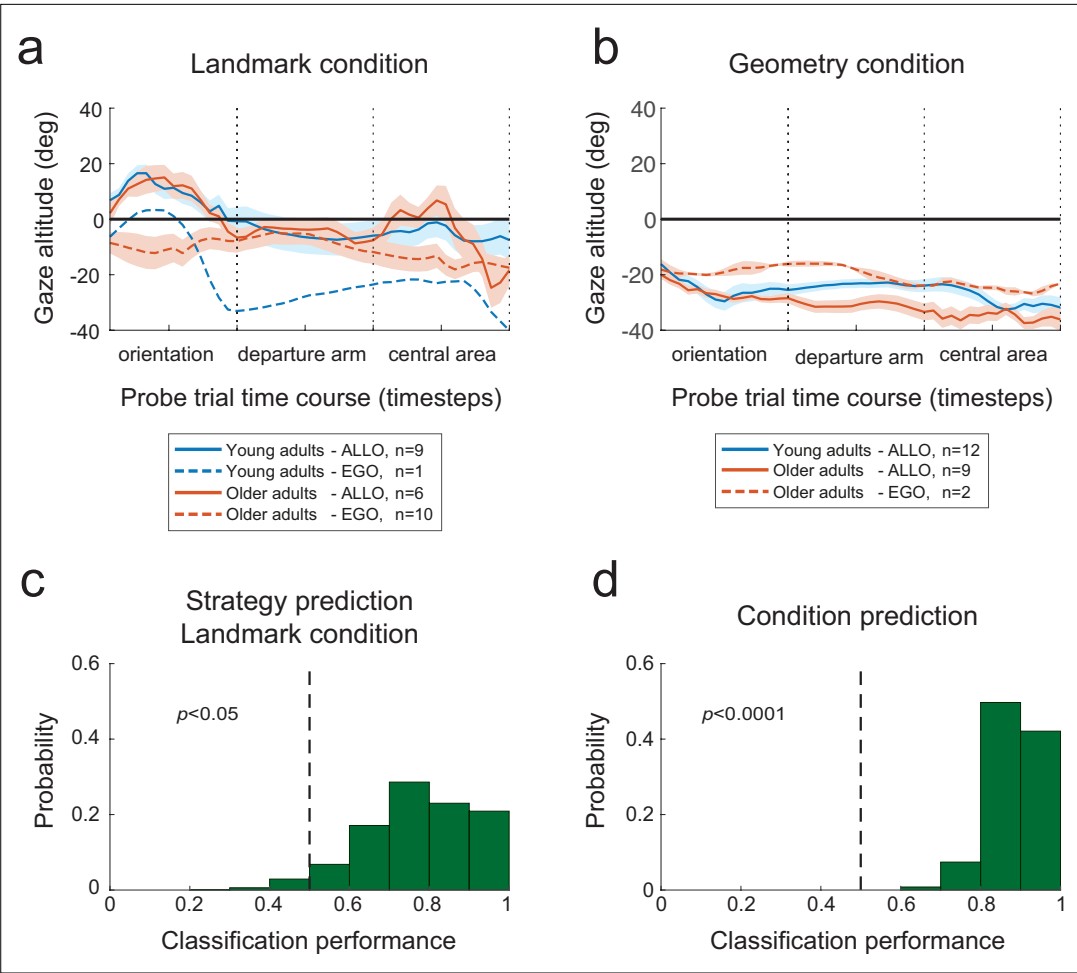

**Figure 8.** Predictive eye-motion statistics. (**a, b**) Evolution of gaze altitude throughout probe trials. In both the landmark (**a**) and geometry (**b**) conditions, the gaze altitude during reorientation differed between allocentric and egocentric navigators. Eye level is denoted by 0. Shaded areas represent the between-subject standard error of the mean. (**c**) Distribution of the proportion of correct single-subject-single-trial predictions of the strategy used to solve the landmark condition of the Y-maze, based on the gaze altitude statistics during reorientation. The dashed vertical line indicates chance-level prediction, that is, the area to the left of the dashed line represents the probability (p-value) that less than half of the subjects in the validation set were correctly classified. (**d**) Prediction performance with respect to the experimental condition, that is, landmark vs. geometry, again on the basis of gaze altitude statistics during reorientation.

3b), and (iii) draw a top-view map of the maze they experienced (*Figure 9—figure supplement 3c and d*). We found that a minority of subjects (15%) made errors in recognizing the maze shape and that all subjects had an intact memory of landmarks (*Figure 9—figure supplement 3e*). In contrast, the association between the remembered landmarks, maze layout, and goal position was problematic as almost half of the subjects (45%) could not place the landmarks in the correct order and even when they did, the landmark array was misaligned either with the maze (45%) or with the goal (65%). Among the 15 subjects who made at least one error on the drawing, 11 (73%) failed to use an allocentric-like strategy in the testing phase. Although performed in a very small sample of subjects, these findings supported our conclusions from the profiling data analyses, in that binding of the landmarks to the cognitive representation of space, rather than the memory of landmarks itself, would underlie the age-related deficits in allocentric-like behavior.

## Discussion

The main finding of this study is that a geometric polarization of the environment removes the egocentric bias in both healthy older adults (>65 yo) and children (~10 yo), and it habilitates their allocentric navigation capabilities. This result provides a strong evidence in support of the hypothesis that age-related allocentric navigation deficits, largely described in the literature (*Colombo et al., 2017*; *Wiener et al., 2013*; *Davis and Weisbeck, 2015*; *Driscoll et al., 2005*; *Rodgers et al., 2012*; *Bohbot et al., 2012*; *van der Ham et al., 2020*; *Newcombe, 2019*; *Nardini et al., 2006*; *Bullens et al., 2010*), can rather be ascribed to an difficulty in using a landmark array for navigation purpose. By means of a Y-maze paradigm enabling natural, physical displacement of subjects in space, we found that children and older navigators are impaired at exhibiting allocentric behaviors when landmarks are the only environmental spatial cues. However, when geometric information is present and informative about the spatial relations between the goal and the subject's position in space, children and older adults become as good allocentric navigators as young subjects.

Through an analysis of locomotor and oculomotor behavior, we showed that learning to orient on the basis of landmark arrays is more difficult than with geometry in children and older adults (independently of the navigation strategy that they put forward in the subsequent probe tests). During learning, all subjects had similar visual exploratory behavior regardless of age (e.g., they all gazed at landmarks for equivalent amount of times). This result suggests that the difficulty in using landmarks for navigation is not merely caused by a lack of visual attention to them. The analysis of neuropsychological and cognitive data as well as post-hoc questionnaires showed that landmark-specific deficits may be linked to difficulties in spatial manipulations of landmark configurations in a navigation-informative way (e.g., mental rotation of a landmark array and inference of self and goal positions in relation to rotated landmarks). These analyses also suggested age-related changes in binding landmark identities to the internal representation of the environment, whereas the memory of landmarks themselves was relatively intact. These conclusions are in line with previous studies showing that estimating and reproducing distances and rotations in a geometrically isotropic environment (i.e., a circular arena) is impaired in older adults (*Mahmood et al., 2009*) and that their performance does not improve when landmarks are provided (*Harris and Wolbers, 2012*). Our data are also in agreement with previous works showing age-related deficits in the retrieval of spatial position (*Jansen et al., 2010*) of landmarks and their temporal order (*Wilkniss et al., 1997*; *Lindenberger and Baltes, 1994*) during route learning. We also show that the observed egocentric bias in aged adults is associated with lower contrast sensitivity to high spatial frequencies. This effect is unlikely to directly cause differences in navigational decisions since landmarks were clearly visible and our participants were screened for normal visual acuity. However, lower visual scores were repeatedly associated with cognitive impairments in large cohort studies (*Lindenberger and Baltes, 1994*; *Roberts and Allen, 2016*; *Naël et al., 2019*), in which impoverished sensory (e.g., visual) input was shown to increase the risk of cognitive decline in aging.

The differential influence of landmark and geometric cues on human navigation suggests a potential dissociation of the related neural processing pathways. If indeed different subnetworks in the brain mediate the processing of different types of cues (*Ramanoël et al., 2022*), our results suggest that the subnetwork dedicated to geometric processing matures earlier in development and it is preserved better in aging. Construction of geometry-based spatial representations could thus represent the basic mode of spatial learning, with efficient binding of landmark to the representations of space developing during primary school years and deteriorating early in aging. The proposed dissociation between brain systems mediating geometric and landmark processing in the brain is not new, and a number of experimental studies and theoretical models addressed this question (*Cheng, 1986*; *Doeller and Burgess, 2008*; *Sheynikhovich et al., 2009*; *Julian et al., 2015*; *Krupic et al., 2015*; *Julian et al., 2016*). However, the age-related aspect of this dissociation is novel. In a recent modeling work, we have proposed that landmark-geometry dissociation may follow a well-established neurophysiological distinction between the dorsal and ventral visual processing streams (*Li et al., 2020*), substantiating earlier proposals related to the role of these pathways in mediating allocentric and egocentric cue representations (*Burgess, 2006*; *Nau et al., 2018*; *Litman et al., 2009*).

The fact that in the geometry condition of the navigation task all age groups had equivalent performance in terms of behavioral and oculomotor measures suggests that children and older adults may not be as bad navigators as previously thought, provided that anisotropic geometric cues are

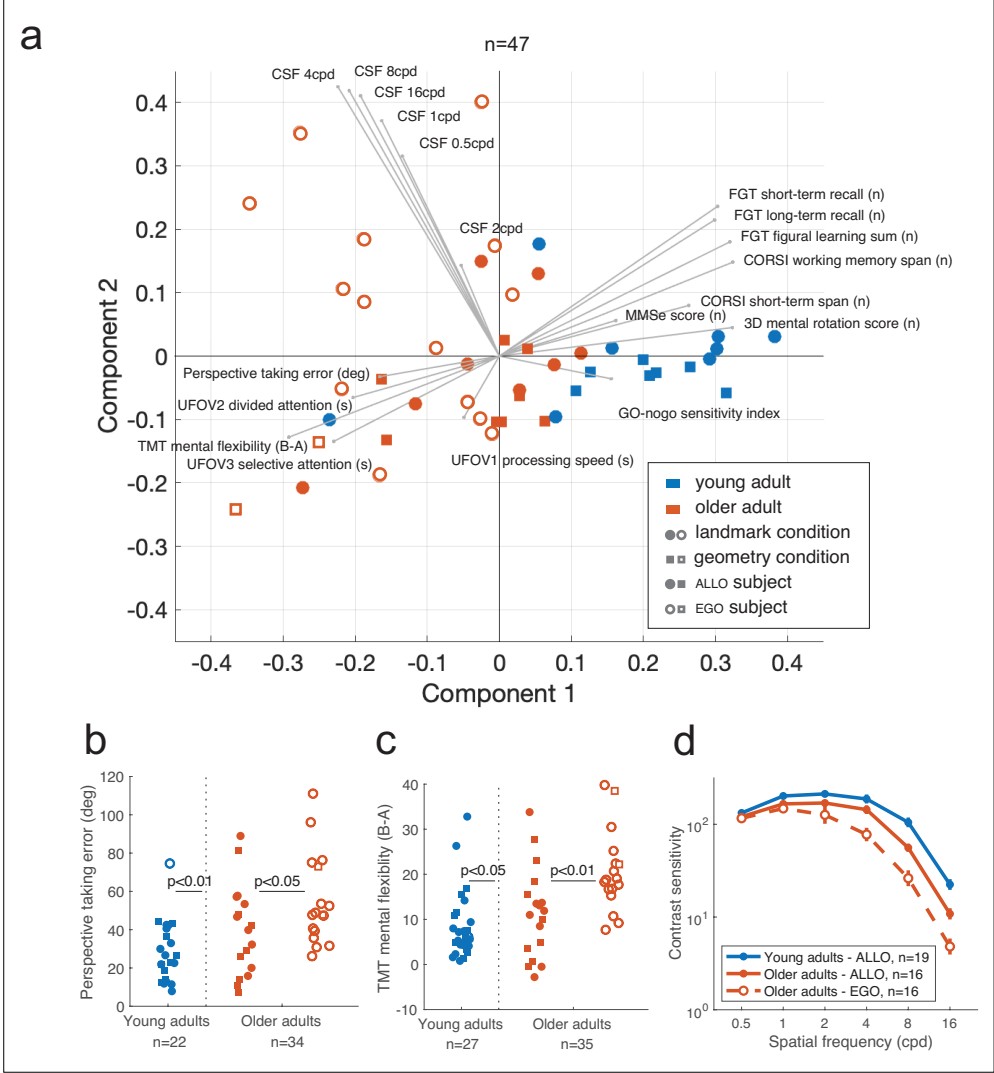

**Figure 9.** Visuo-cognitive multivariate analysis of age-related modulation of spatial behavior. (**a**) Principal component analysis (PCA) across 19 measures of visual, attentional, mnemonic, and spatial reasoning capabilities (see *Supplementary file 3* for test descriptions). Participants could be discriminated based on their age and, within the older population, their strategy preference. PCA was performed on 47 participants for whom we had the complete visuo-cognitive battery. (**b–d**) Scores of perspective taking, TMT mental flexibility, and contrast sensitivity. Error bars in (**d**) represent the standard error of the mean.

The online version of this article includes the following figure supplement(s) for figure 9:

**Figure supplement 1.** Cognitive screening results for adult participants.

**Figure supplement 2.** Visual screening results for adult participants.

**Figure supplement 3.** Results from post-experiment self-reported visuo-spatial memory of landmarks.

available. Confirming previous data from a real-world navigation task (*Bécu et al., 2020*), we show that gazing at the floor contributes to the extraction of geometry-related information about the environment. We put forth an important function of geometry for spatial orientation and navigation across the lifespan, permitting fast learning of the environmental layout, efficient inference of self-position, and the relative goal location in space. Altogether, these findings highlight the necessity to rethink the impact of age on spatial cognition and to reframe the classical allocentric–egocentric dichotomy in terms of landmark-geometry spatial processing and coding. It also calls for rethinking navigational aids and environment architectural designs where older adults and children usually evolve. This article

suggests that emphasizing geometric cues could favor navigation of these populations and reduce their chance of getting lost.

A limitation of our results resides in the fact that we remain blind to the exact neural mechanisms that mediate the anchoring of landmarks to the geometry-based representation of space and how they are affected by aging. Our interpretations thus need to be complemented by neuroimaging investigations, possibly with longitudinal follow-up to reduce potential cohort effects. As a first step toward this issue, experiments should aim at differentiating brain areas implicated in geometry vs. landmark processing in humans, and at characterizing, possibly with longitudinal designs, age-related cortical and subcortical changes potentially linked to the preserved use of allocentric strategies in the presence of geometric cues. Finally, given that older age induces a stronger reliance on the external environment in general (*Lindenberger and Mayr, 2014*) and on visual information in particular (*Agathos et al., 2015*; *Alberts et al., 2019*), results stemming from the coupling of behavioral and neuroimaging will provide a novel insight for rehabilitation solutions for spatial navigation in aging.

## Materials and methods

### Participants

A sample of 79 subjects were enrolled in the virtual reality implementation of the study: 29 children (range: 10–11 y, μ = 10, std = 0.49, 17 females, 12 males), 22 young adults (range: 23–37 y, μ = 28, std = 4.28, 13 females, 9 males), and 28 healthy older adults (range: 67–81 y, μ = 73, std = 3.90, 17 females, 11 males). We used a power analysis to estimate this sample size for our three age groups, with an alpha level of 0.05, a target power of 0.8, and an expected effect size of 0.356, as derived from *Rodgers et al., 2012*. Based on these parameters, a minimal sample size of 77 was estimated in order to reach 0.8 power for our design (*Faul et al., 2007*). For the real-world experiment, we enrolled 17 participants: 9 young adults (range: 22–33 y, μ = 27.39, std = 4, 6 females, 3 males) and 8 healthy older adults (range: 66–78 y, μ = 72.40, std = 4, 4 females, 4 males). All young and older adult participants enrolled in either the virtual reality or the real-world experiment were part of the SilverSight cohort study (*Lagrené, 2019*) at the Paris Vision Institute – Quinze-Vingts National Ophthalmology Hospital. The children were recruited in a primary school in the Paris area. All participants were voluntary, and they (or their parents in the case of children) gave an informed consent for inclusion in the study. All screening and experimental procedures were in accordance with the tenets of the Declaration of Helsinki, and they were approved by the Ethical Committee CPP Ile de France V (ID_RCB 2015-A01094-45, no. CPP: 16122 MSB). Adult participants were included in the study based on the following criteria: (i) corrected visual acuity of at least 7/10 and 5/10 in participants younger and older than 70 yo, respectively; (ii) a Mini-Mental State Examination score of 24 or higher; and (iii) no physical inability in terms of locomoting without assistance. The complete list of inclusion/exclusion criteria used for the SilverSight cohort is described in *Supplementary file 4*. The clinical and functional assessment of the adult participants included ophthalmological screening (e.g., optical coherence tomography, fundus photography), functional visual screening (e.g., visual acuity, visual field extent, contrast sensitivity, attentional field of view), otorhinolaryngological examination (e.g., audiogram, vestibular function), cognitive-neuropsychological assessment (e.g., visuo-spatial memory, mental rotation, executive functions), oculomotor evaluation (e.g., ocular fixation, saccadic control), and a static/dynamic balance assessment. A series of questionnaires were also administered to evaluate the quality of vision with respect to mobility and spatial orientation. In this study, a subset of these multivariate tests were used to avoid sampling biases related to inter-individual characteristics variability and to control for co-factors affecting spatial behavior. Participants habitually wearing far-vision lenses were encouraged to keep their glasses on during the experiment.

### Experimental setup

Adult participants were all tested in the Streetlab experimental platform at the Vision Institute (see *Videos 1–2*). The experiments with children were setup in a school gymnasium. The immersive virtual reality (VR) environment was created using the Unity3D game engine (Unity Technologies), and it was displayed using the HTC VIVE headset equipped with a Tobii Pro VR binocular eye tracker. Participants were equipped with a wireless VR capable backpack system (VR One, MSI). Experiment control and monitoring were performed remotely. This setup allowed the participants to move freely and explore

the immersive virtual space. The head position in the real space was tracked at 30 Hz by two laser emitters placed 9 m away from each other and at a height of 3 m enabling an experimental capture area of approximately 4.0 × 4.0 m. The HTC VIVE headset had a nominal field of view of about 110° through two 1080 × 1200 pixels displays, updated at 90 Hz. The pixel density of the display was about 12 pixels/degree. The Tobii eye-tracker recorded eye movements at a rate of 120 Hz. The equipment used with children was the same as for adults, except that no eye-tracking was performed. The real-world maze was built within the Streetlab experimental platform (*Figure 1—figure supplement 1*). The maze walls were made of black Plexiglas acrylic panels that reflected the visible light while allowing infrared (IR) light to pass through. Body movements were recorded by an opto-electronic motion capture system (10 infrared cameras, model T160) at 120 Hz (VICON Motion Systems Inc). During the experiment, participants wore a tight black suit equipped with 39 infrared reflective markers, following the Vicon Plug-In-Gait model. Marker detection and tracking stability (computed as the position tracking error with respect to a reference measurement without the IR panels) indicated a sub-millimeter accuracy. Participant binocular eye movements were tracked at 60 Hz with SMI eye-tracking glasses (SensoriMotor Instruments).

## Spatial navigation task

Two versions of the Y-maze tasks were used in the immersive virtual reality study: the classical landmark-based equiangular Y-maze (*Figure 1a*) and a new geometrically polarized Y-maze (*Figure 1b*). Both mazes were composed of three corridors, with homogeneously textured walls. In order for all subjects to have to same visual experience of the maze, the wall size was adjusted individually such as to exceed the height of each subject by 10 cm. In the equiangular landmark condition, all corridors were 66 cm wide and 190 cm long. Three distal landmarks were placed 8 m above the maze walls at a distance of 20 m from the maze center: a green star, a blue square, and a red circle, each subtending a visual angle of 10°, if seen from the maze center. In the geometrically polarized condition, all corridors were 66 cm wide and 230 cm long. No distal landmarks were present in this condition. The corridors were longer in the geometric condition than in the landmark one to prevent the subject from seeing the end of the corridor from the departure location. Overall, the maze size in the geometry condition was 1.54 times bigger than in the landmark condition. No shadows were present in the virtual environment and the sky was homogeneous. Subjects were randomly assigned to either the landmark or geometry condition under the constraint of equal gender distribution. In the real-world component of the study, only the landmark-based equiangular Y-maze was implemented (*Figure 1—figure supplement 1*).

## Experimental protocol

The experiment with the virtual maze (in either the landmark of the geometry condition) lasted approximately 30 min per participant. Prior to the experimental session, the headset and the inter-pupillary distance were adjusted for each participant, and a nine-point eye-tracker calibration with head fixed was carried out. The quality of the calibration was verified by using the same nine-point procedure at the beginning, halfway through, and at the end of the experiment. Whenever the mean angular calibration error exceeded 3°, the adjustment/calibration/verification procedure was repeated. Before every trial, the following disorientation procedure was performed: the subject was asked to keep the eyes closed and to hold the hands of the experimenter, while being passively led around the room for approximately 2 min (*Video 1*). To mask uncontrollable sound sources, a non-informative sound was played in the headphones during the whole experiment. To verify the effectiveness of the disorientation procedure, the subject was asked to point to a conspicuous room cue (i.e., a computer, located near the exit of the experimental room). If the pointing error was less than 90°, the entire disorientation procedure was repeated. At the beginning of each trial, the disoriented subject was placed at the departure location (e.g., position A in *Figure 1a*), facing the center of the maze. He/she was instructed not to walk through the virtual walls and not to stand on tiptoes. Before undergoing the two experimental phases (i.e., learning and testing, see below), the subject had to perform three exploration trials (60 s each), starting from each arm of the maze. He/she was explicitly instructed to explore the whole environment. If the subject did not enter one of the arms or did not look up in the direction of the landmarks, he/she was encouraged to do so by the following instruction though the headphones: 'Make sure to explore the whole environment.' At the beginning of each trial of the learning phase, the disoriented subject was placed at the departure position A (*Figure 1a and b*) and

instructed to navigate, as directly as possible, to the goal (dashed C area in *Figure 1a and b*; radius: 0.4 m). The learning phase ended after four consecutive successful trials, defined as the trials in which the subject went directly to the goal without entering the arm B. At the beginning of each trial of the second, testing phase, the disoriented subject was placed in one of the two non-goal arms. The following predefined sequence of starting location areas was used: B, A, A, B, A, B. Each testing trial ended when the subject stopped for at least 5 s at the end of one of the three arms. No reward signal was given during testing trials. Once a trial was ended, the image displayed on the headset faded, the subject was instructed to close the eyes and disorientation began. A questionnaire testing the memory of the experimental environment was administered to a subset of subjects (n = 20) tested in the landmark condition. The protocol used for the experiments conducted in the real-world setting was identical (*Video 2*).

## Data processing

### Navigation measures

Spatial navigation in the virtual maze was assessed separately in the learning and testing phase. To quantify spatial learning, the following navigation variables were measured based on the subjects' trajectories obtained from head tracking data: (i) *trials-to-criterion,* defined as the number of trials until the criterion of four consecutive successful trials was reached (see above for the definition of a successful trial); (ii) *traveled distance,* defined as the length (in meters) of a start-to-goal trajectory; (iii) *escape latency,* calculated as the time (in seconds) necessary to reach the goal zone from the departure location. In the experiments, each trial was separated into an initial *orientation* period, during which the subject opened the eyes and self-located in space, and the subsequent *navigation* period, during the subject physically moved throughout the Y-maze. For analysis purposes, the orientation period was taken as the time interval between the start of image projection on the HMD and locomotion initiation, that is, the moment when the subject exited a virtual circle (radius: 0.3 m) around the departure position. The navigation period was calculated as the time between the start of and the end of locomotion (e.g., upon entrance in the goal area during the learning phase). Therefore, the following time-related variables were used for analyses: (4) *orientation-period duration* and (v) *navigation-period duration*, both measured in seconds. In addition, the (vi) *average speed* was measured as the average instantaneous speed along the trajectory during the navigation period. Given that walking speed is correlated with a person's height (*Samson et al., 2001*), the *normalized speed* (i.e., average speed divided by the height) was used in the statistical comparisons between adults and children. Finally, to assess spatial behavior in the probe trials, (vii) the *time spent in the central area* was calculated as the time (in seconds) spent in the region composed of one third of each of the three arms closest to the center of the maze.

### Navigation strategies

We defined a subject's strategy based on which area the subject first entered on probe trials (areas centered around A, B, and C in *Figure 1a and b*, with a radius of 0.4 m). A subject was classified as belonging to the *allocentric* category if he or she made a majority of allocentric choices in the three probe trials. Otherwise, the subject was classified as *egocentric*. In the landmark version of the virtual experiment, the preferred strategy of one child and one older participant could not be evaluated as they returned several times to the starting position. Those two subjects were excluded in the behavioral and gaze analyses but they were pooled with the egocentric group for the visual and cognitive analyses.

### Eye tracking

In order to measure each subject's gaze vector in the 3D virtual space, head-tracking data were interpolated to synchronize them with the eye-tracker frequency (120 Hz). Once the image was displayed in the HMD, the cyclopean gaze vector was calculated by averaging the data for the left and the right eye. If the signal from one of the eyes was judged too noisy, the recording from the remaining eye was used. The subject's visual focus of attention was computed as the intersection point of the gaze vector with the surface of the virtual walls, the floor, or the sky region, the latter being defined the virtual sphere (radius: 6 m) around the maze center. The *gaze time proportion* was calculated as the proportion of time that the gaze focused on either the walls, the floor, or the sky region, normalized

by the total duration of the period considered (see below). There were missing data (young adults: 22%; older adults: 27%) due to blinks or loss of the pupil tracking. These were not considered when computing the gaze time proportion. Fixations and saccades were not separated for the analyses. For statistical comparisons, the gaze time proportion was computed for each trial and averaged across subjects and/or experimental trials. To visualize the distribution of the gaze foci as a function of trial duration, the gaze time proportion was computed over a window of 1 s sliding across the trial time separated into 50 time bins (orientation period: 15 bins; navigation period: 35 bins). Finally, to visualize the spatial distribution of the gaze foci in a particular experimental condition, they were accumulated from all trials and all subjects considered, for a particular region (e.g., sky, floor), and represented by heatmaps, normalized to have the maximal value of 1.

### Predictive model based on eye-motion statistics

A generalized linear (logistic) regression model was trained to predict the navigation strategy (i.e., allocentric vs. egocentric) as well as the experimental condition (i.e., landmarks vs. geometry) based on the average altitude of the gaze relative to eye level (in degrees) recorded during the orientation period of probe trials. Gaze altitude was expressed as the elevation (in degrees) of the gaze vector relative to a horizontal plan passing though the eye height. To assess the regression performance, we used two cross-validation procedures. First, 25% hold-out validation, in which the model was trained using 75% of the subjects' data and then tested on the remaining subjects' data. The same procedure was repeated 1000 times to estimate the distribution of the proportion of the correct responses. The p-value reported in the figures corresponds to the probability that the prediction was correct for less than half of the subjects. Second, leave-one-out validation, in which the model was trained on all but 1 subject, whose data were used for testing. The procedure was repeated for all subjects, and the number of correctly classified cases was reported. Strategy prediction was assessed only for the landmark condition since there were only two subjects that made egocentric choices in the geometry condition.

### Scoring self-reported outcomes and map drawings

At the end of the experimental session in the 3D virtual environment, a subset of participants were asked to recall the shape of the maze (out of three choices) as well as of the three landmarks (out of a set of six, with three distractors). They were also asked to sketch a top-view drawing of the maze, indicating the landmark array and the goal location. A binary score (correct/incorrect) was used to quantify performance in recalling the shape of the maze and of the landmarks. Three binary scores were used to evaluate the map sketches. They assessed whether the order of landmarks was correct (independently of their position in space with respect to the maze), whether the placement of landmarks relative to the maze was correct (i.e., if the landmarks were drawn in-between the arms and not at their ends), independently of the landmark order, and whether the goal zone was placed in the correct arm and relative to the landmarks.

## Statistical analyses

Student's t-tests and ANOVAs were used for statistical comparisons on continuous data that passed normality and homoscedasticity tests. The normality was verified by the Lilliefors normality test and by the visual inspection of Q-Q plots. Box–Cox transformations were used on this data. ANOVAs were complemented by Bayesian ANOVAs for null testing (*Rouder et al., 2009*) performed with the Bayes Factor package in R. Nonparametric tests were used for ordinal data and when normality could not be achieved. Mann–Whitney U test was used to compare independent groups. When samples were large, we used the z approximation to calculate the Cohen's r, as an estimation of the effect size with $r = z/\sqrt{N}$. We also specify the global n for the two groups compared. Cohen's guidelines for r are that a large effect is 0.5, a medium effect is 0.3, and a small effect is 0.1. Wilcoxon signed-rank W statistic was used to test whether the difference between pairs of observations differed from a given median (in our case, 0). We complemented nonparametric tests with a Bayesian rank-based testing method described in *van Doorn et al., 2020*. Alpha level for statistical significance was set to 0.05. Corrections for multiple comparisons are described in the text. To assess statistical interactions between age and condition in the Y-maze task, a logistic regression model (generalized linear model with binomial response and logit link) was fit to the data.

Interactions in this model were analyzed using marginal effects framework (*Mize, 2019*) implemented by the 'marginaleffects' R package. In the text, we report second differences (Δ, the effect of change from landmark to geometry condition on the difference between age groups) and their p-values. Alpha level is 0.05 is used.

## Acknowledgements

We thank Saddek Mohand-Said of the Clinical Investigation Centre of the Quinze-Vingts Hospital, Paris for medical supervision during clinical screening of participants. We also thank Karine Lagrené, Sonia Combariza, and Jérôme Gillet from the Aging in Vision and Action laboratory at Vision Institute for helping in enrolling/profiling the participants. Finally, the authors wish to thank Emanuel Gutman, Johan Lebrun of the Streetlab team for technical support in setting up the experiments in the Streetlab platform. ANR-Essilor SilverSight Chair grant ANR-14-CHIN-0001, ANR-Essilor SilverSight Chair grant ANR-18-CHIN-0002, LabEx LIFESENSES grant ANR-10-LABX-65, IHU FOReSIGHT grant ANR-18-IAHU-01.

## Additional information

### Funding

| Funder | Grant reference number | Author |
| --- | --- | --- |
| ANR | ANR-14-CHIN-0001 ANR-14-CHIN-0002 | Angelo Arleo |
| ANR | Labex LifeSenses ANR-10-LABX-65 | José-Alain Sahel Angelo Arleo |
| ANR | IHU FOReSIGHT grant ANR-18-IAHU-01 | José-Alain Sahel Angelo Arleo |

The funders had no role in study design, data collection and interpretation, or the decision to submit the work for publication.

### Author contributions

Marcia Bécu, Conceptualization, Resources, Data curation, Software, Formal analysis, Supervision, Validation, Investigation, Visualization, Methodology, Writing – original draft, Project administration, Writing – review and editing; Denis Sheynikhovich, Conceptualization, Supervision, Funding acquisition, Methodology, Writing – original draft, Writing – review and editing; Stephen Ramanoël, Conceptualization, Visualization, Methodology; Guillaume Tatur, Data curation, Software, Investigation, Visualization, Methodology; Anthony Ozier-Lafontaine, Data curation, Software, Formal analysis, Investigation, Visualization, Methodology; Colas N Authié, Validation, Investigation, Methodology; José-Alain Sahel, Supervision, Funding acquisition, Project administration, Writing – review and editing; Angelo Arleo, Conceptualization, Supervision, Funding acquisition, Methodology, Writing – original draft, Project administration, Writing – review and editing

### Author ORCIDs

Marcia Bécu ⬤ http://orcid.org/0000-0003-4564-1023
Denis Sheynikhovich ⬤ http://orcid.org/0000-0001-7737-8907
Stephen Ramanoël ⬤ http://orcid.org/0000-0003-4735-1097

### Ethics

Human subjects: All participants were voluntary and they (or their parents in the case of children) gave an informed consent for inclusion in the study. All screening and experimental procedures were in accordance with the tenets of the Declaration of Helsinki, and they were approved by the Ethical Committee CPP Ile de France V (ID_RCB 2015-A01094-45, No. CPP: 16122 MSB).

### Decision letter and Author response

Decision letter https://doi.org/10.7554/eLife.81318.sa1
Author response https://doi.org/10.7554/eLife.81318.sa2

# Additional files

## Supplementary files

• Supplementary file 1. Number of observations, mean, and standard deviation of age in different age groups.

• Supplementary file 2. Number of observations, mean, and standard deviation of age in different age groups for the experiments in real-world setting.

• Supplementary file 3. List of visual and cognitive tests performed by a subset of our adult participants.

• Supplementary file 4. Inclusion/exclusion criteria used for the SilverSight cohort (adult participants).

• MDAR checklist

• Source data 1. Participants' demographics.

## Data availability

All data and code used in the analyses are available as an Open Science Framework deposit, accessible at https://osf.io/zhrk4.

The following dataset was generated:

| Author(s) | Year | Dataset title | Dataset URL | Database and Identifier |
|---|---|---|---|---|
| Bécu M | 2022 | Evolution of landmark-based spatial navigation across the human lifespan | https://osf.io/zhrk4 | Open Science Framework, zhrk4 |

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
