## [Editor Report]

The findings in the article show that when provided with geometric cues, rather than landmark cues, older adults and children no longer show selective difficulty with learning spatial layouts allocentrically. This important and compelling finding challenges decades of work suggesting that older adults have a selective allocentric navigation deficit. Instead, the findings suggest that older adults may have perceptual issues related to processing and integrating landmarks.

---

## [Decision Letter]

**Decision letter after peer review:**

Thank you for submitting your article "Evolution of landmark-based spatial navigation across the human lifespan" for consideration by *eLife*. Your article has been reviewed by 2 peer reviewers, and the evaluation has been overseen by a Reviewing Editor and Chris Baker as the Senior Editor. The reviewers have opted to remain anonymous.

As editor, I have also carefully reviewed the paper and the reviewer comments. Overall, I think the paper makes a novel and important point potentially worthy of publication. Specifically, the manuscript nicely converges on the idea that past literature arguing for allocentric spatial deficits in older adults is flawed. Instead, the results in the current paper make the point that such differences in fact arise from differences in visual landmark processing in older adults and not due to a deficit in allocentric representation per se. I believe that the basic results supporting this idea are robust and the experimental design sound and well-reasoned in terms of addressing the core issue of landmark vs. geometrical processing differences and their relationship to allocentric vs. egocentric processing.

Nonetheless, both reviewers, particularly reviewer #1, identified serious statistical and analytic choices that limit the impact of the paper. As such, I am rejecting the current version of the paper and asking you to consider carefully revising your manuscript based on the feedback below.

The major issues I see here are statistical and in terms of the presentation. Perhaps most important, a null difference, particularly in a small sample, is not evidence of a lack of a difference. Bayes null testing, where possible, should be used (Rouder et al. 2009), and all null statements should be treated with greater caution. In addition, if statements are made about differences in one condition but not another, interaction effects should be tested for and identified as significant; otherwise, there could simply be differences in variance that are not accounted for (Nieuwenhuis et al. 2011). The areas involving eye tracking and neuropsychological tests were somewhat hard to follow and the number of subjects in these subsamples difficult to determine. One possibility could involve increasing the sample size for the neuropsychological correlations/PCA as it is currently significantly under-powered. Statistical power should be reported when possible to ensure that these effects can be considered robust. Lastly, the choice of the Mann-Whitney U test, while justified with non-parametric data, should be handled more carefully. In the very least, correction for multiple comparisons should be applied and/or non-parametric tests used that allow for testing of interaction effects. For categorical data (maze choice), the authors could consider a multinomial logistic regression. Also, where possible, if the data are continuous and fit the correct parametric assumptions, ANOVAs should be applied.

*Reviewer #1 (Recommendations for the authors):*

I have several comments to the authors:

1. One difference between the mazes is that the geometry condition maze is 54% longer than the landmark condition maze. I don't think this can cause the observed effects, but this should be mentioned in the limitations section and not just hidden in the methods. It might also be mentioned in Figure 3 legend as this affects its interpretation (e.g. larger walking distances in the geometry compared to landmarks maze in all groups.)

2. The potential dissociation between geometry-based and landmark-based processing is a strength of this study – although this has been discussed in the literature, people still think on "allocentric mapping" as one system where the cognitive map is anchored to either geometry or landmarks. The study design very elegantly dissociates these elements, and I think this is important and can be emphasized more strongly (even in the abstract). Also, with regards to previous literature, besides the influential Doeller and Burgess 2008 paper on dissociation between geometric and landmark-based navigation that is cited, the studies by Julian et al. PNAS 2015 (dissociation between feature/landmark-based context retrieval and geometry-based heading retrieval), Krupic et al. Nature 2015 (geometry-based coding irrespective of distal cue location), and Julian et al. Curr Biol 2016 (impairment in geometry-based location coding without impairment in landmark-based location coding) might be relevant.

3. The Results section is very long and detailed, and requires substantial reader attention to go through all of it and understand the findings. Subheadings to the different sections detailing the main effects could be very beneficial for readability and to enable getting a grasp of the findings without reading all sections fully.

4. The authors sometimes refer to the maze as a "radial maze" and sometimes as a "Y maze". I think Y-maze is more appropriate here and should be used consistently (including in the abstract) to avoid confusion.

5. The authors sometimes use the term "landmark processing" which might be confusing – since the problem doesn't seem to be with processing the landmarks (i.e. noticing/attending to them and being able to recognize them), but with using them to navigate. I would change the terminology to "using landmarks during navigation" or something similar.

6. The age (mean+-SD) should be stated at the start of the results as this is a very important issue and shouldn't be mentioned only in the methods.

7. Figure 5 is unclear without reference to the text: In panel A, there is no marking of the average gaze time to the sky in the learning trials. (2) In panel B, the original and probe starting locations are not marked. (3) In panel C, the color scale emphasizes the learning effect but makes it difficult to see what participants focus on during navigation – it might be better to separate the learning and navigation phases, and modify the scale of the navigation period graph to better emphasize the effects.

8. The authors may optionally wish to speculate on the potential impact of their study in real life – how navigational aids or environments can be designed to alleviate the problems faced by older adults or children during navigation.

*Reviewer #2 (Recommendations for the authors):*

1. The authors report significant main effects of age across several outcome measures in the landmark condition. Most notably, children and older adults are more likely to engage in an 'egocentric' strategy during probe trials. Similar age effects are largely absent in the geometric condition; children, young adults, and older adults are equally likely to engage in an 'allocentric' strategy during probe trials and generally do not differ across any of the other outcome measures. It isn't clear, however, whether the authors performed any analyses necessary to identify a significant age group x condition interaction, which is necessary to determine whether the availability of geometric cues truly moderates the effect of age on navigation. To put it another way, simply showing a significant main effect of age in one condition and a null effect in the other does not in itself indicate that the magnitude of the age differences were moderated by the respective conditions. Given the data presented in Figure 2, I suspect this will be the case (with sufficient power, at least), but the results of formal interaction analyses should be reported.

2. The authors do a deep dive into the eye tracking analyses, which is informative but often difficult to follow. The results often switch back and forth between describing results of between-condition comparisons (i.e., landmark vs geometric) and within-landmark comparisons (i.e. allocentric vs. egocentric). It also wasn't always clear whether data from the VR (landmark and geometric) and real-world (landmark only) conditions were collapsed when describing the principal analyses. This seems particularly relevant when considering the use of different systems to obtain eye tracking data. Were any measures taken to compare the reliability and/or precision of gaze data measured by the respective systems?

3. During the orientation phase on probe trials in the landmark condition, young and older allocentric navigators tended to orient towards the star. During the subsequent navigation phase, older allocentric navigators showed a greater tendency to orient towards the red circle, which the authors suggest may reflect a cue-based view matching strategy. By contrast, young adults continue to orient towards the star during navigation, which is interpreted as reflecting a cognitive map-based strategy. Curiously, older egocentric navigators exhibit viewing patterns similar to that observed in younger allocentric navigators. These results appear to be purely based on a qualitative interpretation of the heatmaps presented in Figure 5C. Were there any formal statistics on gaze dwell-time to confirm these ostensive age differences in the evolution of viewing patterns? The authors do report quantitative age differences in orientation latencies, amount of time spent in the central maze area, and escape latencies to support these interpretations, but the link between these measures seems highly speculative.

4. The authors performed a classifier analysis to determine whether gaze altitude during orientation (that is, viewing the floor or sky) could predict subsequent navigation strategies. How was altitude quantified? Was it based on mean angle/degrees computed across the entire orientation epoch? Likewise, Figure 7A and 7B suggest that there was substantially more variability in gaze altitude during the orientation phase in the landmark condition compared to the geometric condition (both between- and, perhaps more importantly, within-groups). Can the authors discuss what this difference in gaze variability between conditions might mean in terms of interpreting the classifier analysis, and whether it represents a potential confound?

5. A subset of participants completed a battery of 19 visual and neurocognitive tests. Among older adults, those that showed a bias for egocentric navigation also tended to perform worse on tests of perspective taking, mental flexibility, and contrast sensitivity. Did the authors correct for multiple comparisons? Several of these effects do not appear to be particularly strong, and since these tests were only performed in a subset of participants, the broader implications of these results are difficult to determine.

6. In Supplementary Figure 4, the authors note that trial-to-criterion, travelled distance, and escape latency did not differ between the landmark and geometric conditions in young adults. The authors argue that this speaks to the comparable levels of difficulty between the two conditions. Can the authors elaborate on this? I don't follow the rationale that null effects in young adults is a sensitive measure of task complexity experienced by children and/or older adults.

7. Line 410: What was the rationale for adopting different visual acuity inclusion criteria for young (7/10) and older (5/10) adults?

8. Why were neuropsychological, visual acuity, and post-task questionnaires only collected in a subset of participants?

Some other issues the editor (Ekstrom) noted:

1) "Egocentric strategies rely on spatial codes anchored on the subject's body, whereas allocentric strategies are grounded on representations that are independent from the subject's position and orientation, akin to a topographic map (4)."

It should be noted that egocentric processing also can involve (and often does involve) visual snapshots of the environment, see for example (Waller and Hodgson 2006).

2) "The study of age-related changes in human navigation has added a new temporal dimension to this research domain by investigating the evolution of spatial learning and wayfinding behavior across the lifespan."

As this is a cross-sectional study (old vs. young vs. children) and not longitudinal, it seems difficult to rule out cohort effects. Perhaps instead phrase as age-related differences.

3) "Hence, an alternative explanation consistent with the literature is that the widely-accepted hypothesis of age-related allocentric deficit may in fact reflect a landmark-processing impairment."

It is unclear if what is being looked at is a "deficit" or a difference in processing/strategy.

4) "We sought to test these hypotheses by employing a radial-maze experimental paradigm, traditionally used to dissociate egocentric and allocentric navigation in rodents (27,28) and humans (15)."

From the way things are written, it is difficult to tell which findings came from the real-world maze and the virtual maze. This should be made clearer upfront and at all points in the results, this distinction should be clearer.

5) "We sought to test these hypotheses by employing a radial-maze experimental paradigm, traditionally used to dissociate egocentric and allocentric navigation in rodents (27,28) and humans (15).

The sample size varies somewhat throughout the manuscript. It should be made clear throughout exactly what the N is in each comparison.

6) "Older adults required a higher number of trials than young adults to reach the learning criterion of 4 consecutive successful trials (Figure 3A; older vs. young adults: U=287.5, p<0.05).

It is unclear what test is being conducted here and what the degrees of freedom are.

7) "A subset of the participants in the virtual experiment underwent a complete battery of visual and neurocognitive screenings, resulting in 19 measurements per subject

How many subjects were tested here?

References

Nieuwenhuis S, Forstmann BU, Wagenmakers EJ. 2011. Erroneous analyses of interactions in neuroscience: a problem of significance. Nature neuroscience 14: 1105-7

Rouder JN, Speckman PL, Sun DC, Morey RD, Iverson G. 2009. Bayesian t tests for accepting and rejecting the null hypothesis. Psychon B Rev 16: 225-37

Waller D, Hodgson E. 2006. Transient and enduring spatial representations under disorientation and self-rotation. J Exp Psychol Learn Mem Cogn 32: 867-82

---

## [Author Response]

As editor, I have also carefully reviewed the paper and the reviewer comments. Overall, I think the paper makes a novel and important point potentially worthy of publication. Specifically, the manuscript nicely converges on the idea that past literature arguing for allocentric spatial deficits in older adults is flawed. Instead, the results in the current paper make the point that such differences in fact arise from differences in visual landmark processing in older adults and not due to a deficit in allocentric representation per se. I believe that the basic results supporting this idea are robust and the experimental design sound and well-reasoned in terms of addressing the core issue of landmark vs. geometrical processing differences and their relationship to allocentric vs. egocentric processing.Nonetheless, both reviewers, particularly reviewer #1, identified serious statistical and analytic choices that limit the impact of the paper. As such, I am rejecting the current version of the paper and asking you to consider carefully revising your manuscript based on the feedback below.The major issues I see here are statistical and in terms of the presentation. Perhaps most important, a null difference, particularly in a small sample, is not evidence of a lack of a difference. Bayes null testing, where possible, should be used (Rouder et al. 2009), and all null statements should be treated with greater caution. In addition, if statements are made about differences in one condition but not another, interaction effects should be tested for and identified as significant; otherwise, there could simply be differences in variance that are not accounted for (Nieuwenhuis et al. 2011). The areas involving eye tracking and neuropsychological tests were somewhat hard to follow and the number of subjects in these subsamples difficult to determine. One possibility could involve increasing the sample size for the neuropsychological correlations/PCA as it is currently significantly under-powered. Statistical power should be reported when possible to ensure that these effects can be considered robust. Lastly, the choice of the Mann-Whitney U test, while justified with non-parametric data, should be handled more carefully. In the very least, correction for multiple comparisons should be applied and/or non-parametric tests used that allow for testing of interaction effects. For categorical data (maze choice), the authors could consider a multinomial logistic regression. Also, where possible, if the data are continuous and fit the correct parametric assumptions, ANOVAs should be applied.

General comment from the authors:

We have revised the paper by taking into consideration the comments of the editor and reviewers, notably in terms of statistics. Specifically, we did the following:

– We have replaced Mann-Whitney U test by two-ways ANOVAs in order to assess main effects and interactions, when normality could be achieved with a boxcox transformation. This led to a substantial change in the result section. Our main conclusions from the previous version of the paper are maintained, while some have been downplayed.

– We have added Bayes factor to classical p-value ANOVAs presentation for a better readout of the effect strength favoring the alternative or the null hypothesis.

– We have added interaction analysis using a binomial logistic regression model for the categorical data in Figure 2, as suggested by the editors. The interactions in this model were analyzed using marginal effects framework (implemented in the “marginaleffects” package in R).

– We have collected additional cognitive and visual data and pooled participants from the virtual and real experiments in order to increase the statistical power of the principal component analysis (PCA).

– We have added headings to the Results section to improve readability thereof.

– Non-parametric testing has been maintained in the case of non-normal data. It was the case when comparing gazing time proportion (e.g., time spent gazing at landmarks or the floor, in Figure 4) or in comparison with small samples (e.g., analysis comparing older adults using allocentric vs. egocentric strategies in Figure 5). We backed up the classical tests with a non-parametric approach from van Doorn et al. (see added reference) in order for the reader to understand how data support either the null hypothesis or its alternative.

Reviewer #1 (Recommendations for the authors):I have several comments to the authors:1. One difference between the mazes is that the geometry condition maze is 54% longer than the landmark condition maze. I don't think this can cause the observed effects, but this should be mentioned in the limitations section and not just hidden in the methods. It might also be mentioned in Figure 3 legend as this affects its interpretation (e.g. larger walking distances in the geometry compared to landmarks maze in all groups.)

We agree that this information, also rendered on Figure 1B, should have been stated clearly. We now do so in both the caption of Figure 1B and in the Results section, line 166, to explain why participants travelled longer distances in the geometric maze.

2. The potential dissociation between geometry-based and landmark-based processing is a strength of this study – although this has been discussed in the literature, people still think on "allocentric mapping" as one system where the cognitive map is anchored to either geometry or landmarks. The study design very elegantly dissociates these elements, and I think this is important and can be emphasized more strongly (even in the abstract). Also, with regards to previous literature, besides the influential Doeller and Burgess 2008 paper on dissociation between geometric and landmark-based navigation that is cited, the studies by Julian et al. PNAS 2015 (dissociation between feature/landmark-based context retrieval and geometry-based heading retrieval), Krupic et al. Nature 2015 (geometry-based coding irrespective of distal cue location), and Julian et al. Curr Biol 2016 (impairment in geometry-based location coding without impairment in landmark-based location coding) might be relevant.

We agree with the reviewer on the interpretation of two dissociable systems. This point has been strengthened in the revised abstract. We also thank the reviewer for the references, which were cited in the dedicated discussion paragraph.

3. The Results section is very long and detailed, and requires substantial reader attention to go through all of it and understand the findings. Subheadings to the different sections detailing the main effects could be very beneficial for readability and to enable getting a grasp of the findings without reading all sections fully.

We have added subheadings to facilitate reading.

4. The authors sometimes refer to the maze as a "radial maze" and sometimes as a "Y maze". I think Y-maze is more appropriate here and should be used consistently (including in the abstract) to avoid confusion.

We have updated the text accordingly.

5. The authors sometimes use the term "landmark processing" which might be confusing – since the problem doesn't seem to be with processing the landmarks (i.e. noticing/attending to them and being able to recognize them), but with using them to navigate. I would change the terminology to "using landmarks during navigation" or something similar.

This is an interesting language issue raised by the reviewer. In our definition, visuo-spatial processing pertains to the ability to perceive, manipulate, and think about visual patterns in space, including the ability to determine where objects are in space relative to oneself and others. Yet, we agree that “landmark/spatial cues processing” might be misleading in the sense that it relates to both lower-level skills (perceiving) and higher-level skills (spatial reasoning and positioning). We have revised the text to clarity to what extent accounting for the processing of landmarks in a navigation context is key to explain age-related differences (e.g., lines 386 and 409).

6. The age (mean+-SD) should be stated at the start of the results as this is a very important issue and shouldn't be mentioned only in the methods.

We have updated the text accordingly (line 78).

7. Figure 5 is unclear without reference to the text: In panel A, there is no marking of the average gaze time to the sky in the learning trials. (2) In panel B, the original and probe starting locations are not marked. (3) In panel C, the color scale emphasizes the learning effect but makes it difficult to see what participants focus on during navigation – it might be better to separate the learning and navigation phases, and modify the scale of the navigation period graph to better emphasize the effects.

Figure 5 is dedicated to probe trials only, which is now indicated more clearly by subheadings. This is why we choose to not show this data for the learning trials (which can be seen on Figure 4F). We nevertheless think that comparing learning (4F) and probe (5A) trials is interesting as it shows how people that use an allocentric strategy adapt their behavior to the new starting location. We have modified the caption of Figure 5A in order to better explain this point and the link to the preceding figure (Figure 4). We have also modified Figure 5B to indicate the starting location of the probe trials only, to strengthen the fact that this figure is about probe trials, and to avoid confusion with preceding (Figure 4).

Concerning the panel C (heatmaps), the variable represented here is the proportion of time participants gazed at different elements of the environment during a given time window. We average this across subjects, without any normalization so the data are directly readable (i.e., a data of 0.5 means that on average participant spent 50% time gazing at the element on a given time window). We think that the use of a normalization or different colormap for the two periods would have removed this direct relation.

We nevertheless agree that the navigation period was less readable, since participants did not gaze at landmark so much as in the orientation period. We have now used a different the colormap (a more discretized one) to increase the readability of the gaze behavior map during the navigation period.

8. The authors may optionally wish to speculate on the potential impact of their study in real life – how navigational aids or environments can be designed to alleviate the problems faced by older adults or children during navigation.

We thank the reviewer for the suggestion. We have added this interesting point (lines 427-430).

Reviewer #2 (Recommendations for the authors):1. The authors report significant main effects of age across several outcome measures in the landmark condition. Most notably, children and older adults are more likely to engage in an 'egocentric' strategy during probe trials. Similar age effects are largely absent in the geometric condition; children, young adults, and older adults are equally likely to engage in an 'allocentric' strategy during probe trials and generally do not differ across any of the other outcome measures. It isn't clear, however, whether the authors performed any analyses necessary to identify a significant age group x condition interaction, which is necessary to determine whether the availability of geometric cues truly moderates the effect of age on navigation. To put it another way, simply showing a significant main effect of age in one condition and a null effect in the other does not in itself indicate that the magnitude of the age differences were moderated by the respective conditions. Given the data presented in Figure 2, I suspect this will be the case (with sufficient power, at least), but the results of formal interaction analyses should be reported.

We are thankful to the reviewer for pointing out this important flaw in our statistical analyses. To correct it, we have fit a logistic regression model to the data as suggested by the editors. We used marginal effects framework (Mize et al. 2019, now cited in the manuscript), implemented in the R package *marginaleffects*, in order to assess the statistical significance of interactions in this model. We now report the results of this analysis on page 9 of the revised manuscript, confirming a significant interaction between age and condition in our data. A description of this analysis has been added to the Methods (p. 41).

2. The authors do a deep dive into the eye tracking analyses, which is informative but often difficult to follow. The results often switch back and forth between describing results of between-condition comparisons (i.e., landmark vs geometric) and within-landmark comparisons (i.e. allocentric vs. egocentric). It also wasn't always clear whether data from the VR (landmark and geometric) and real-world (landmark only) conditions were collapsed when describing the principal analyses. This seems particularly relevant when considering the use of different systems to obtain eye tracking data. Were any measures taken to compare the reliability and/or precision of gaze data measured by the respective systems?

We understand that the section devoted to eye movements was particularly long and difficult to follow. We have added subheadings to the Result section, in the form of a compact sentence specifying the main subsection message. We believe that this will significantly facilitate the reading of the section.

We have indeed recorded eye movements in the real-world experiment, but the data presented in this manuscript come only from the VR-based eye-tracker. Under no circumstance have we collapsed the data from the two eye-trackers. We have clarified this in lines 136 and 198. Although we have found gaze patterns in real-world condition to be coherent with the results in VR (see Author response image 1 showing gazing time proportion in the real-world replica of the landmark condition), we think that such VR/real-world comparison is beyond the scope of the paper and it would add the already cluttered result section.

**Author response image 1. sa2fig1:** 

3. During the orientation phase on probe trials in the landmark condition, young and older allocentric navigators tended to orient towards the star. During the subsequent navigation phase, older allocentric navigators showed a greater tendency to orient towards the red circle, which the authors suggest may reflect a cue-based view matching strategy. By contrast, young adults continue to orient towards the star during navigation, which is interpreted as reflecting a cognitive map-based strategy. Curiously, older egocentric navigators exhibit viewing patterns similar to that observed in younger allocentric navigators. These results appear to be purely based on a qualitative interpretation of the heatmaps presented in Figure 5C. Were there any formal statistics on gaze dwell-time to confirm these ostensive age differences in the evolution of viewing patterns? The authors do report quantitative age differences in orientation latencies, amount of time spent in the central maze area, and escape latencies to support these interpretations, but the link between these measures seems highly speculative.

We have performed the statistical analysis of the time proportion of gazing at the different landmarks during the first probe trial (more precisely, across different periods of the first probe trial). It has appeared that older allocentric navigators spend higher amount of time gazing at the circle than their younger counterparts (see the median in the new Supplementary Figure 5C). Because the statistical comparison of these data did not reach the level of significance, the revised text mentions that the data point in the direction of a view-matching in older adults. We have also downplayed our interpretation of a view-matching strategy in older adults.

4. The authors performed a classifier analysis to determine whether gaze altitude during orientation (that is, viewing the floor or sky) could predict subsequent navigation strategies. How was altitude quantified? Was it based on mean angle/degrees computed across the entire orientation epoch? Likewise, Figure 7A and 7B suggest that there was substantially more variability in gaze altitude during the orientation phase in the landmark condition compared to the geometric condition (both between- and, perhaps more importantly, within-groups). Can the authors discuss what this difference in gaze variability between conditions might mean in terms of interpreting the classifier analysis, and whether it represents a potential confound?

Gaze altitude was expressed as the elevation (in degrees) of the gaze vector relative to the horizontal plan passing though the eye height. Therefore, zero corresponded to participant eye height, and values were either positive or negative when the participant gazed upwards or downwards, respectively. The performance of the classifier was assessed on altitude data averaged over the orientation epoch. We have added a more detailed explanation of how these data were calculated in the revised manuscript (line 596).

When predicting the condition each participant was assigned to, the classification performance was 88% (on average over 1000 iterations). The average performance was 97% in the geometry group and 81% in the landmark group.

When predicting the strategy in the landmark group, the classifier performance was 79%, with 88% and 61% correct classification for allocentric and egocentric navigators.

We think the lower predictability in the landmark group, and specifically for older adults egocentric subgroup, is likely to reflect the higher gaze altitude variability observed. We have now pointed out this possibility in lines 323-330.

5. A subset of participants completed a battery of 19 visual and neurocognitive tests. Among older adults, those that showed a bias for egocentric navigation also tended to perform worse on tests of perspective taking, mental flexibility, and contrast sensitivity. Did the authors correct for multiple comparisons? Several of these effects do not appear to be particularly strong, and since these tests were only performed in a subset of participants, the broader implications of these results are difficult to determine.

The complete set of 19 measurements was not conducted for all participants included in the study at the moment of the navigation experiments. In response to the concern of the editor and reviewers, we have now added as many subjects as possible for these visual and cognitive analyses. In addition to this, we have also pooled the visuo-cognitive data from participants that underwent the virtual and real-world experiments, reaching a sample of 64 for this second version of the manuscript. With this larger sample we have performed more sophisticated statistical analysis, and we have substantially modified Figure 8. We have specified in the revised text when correction for multiple comparison was applied.

Unfortunately, we were unable to get the screening data from all participants. This is due to the fact that some participants drop out the SilverSight cohort before finishing the 19 measurements, which were performed during different visits to the lab. Drop-out occurred for several reasons that are usual for aging cohort study populations (e.g., relocation, unwillingness to further participate, death).

6. In Supplementary Figure 4, the authors note that trial-to-criterion, travelled distance, and escape latency did not differ between the landmark and geometric conditions in young adults. The authors argue that this speaks to the comparable levels of difficulty between the two conditions. Can the authors elaborate on this? I don't follow the rationale that null effects in young adults is a sensitive measure of task complexity experienced by children and/or older adults.

We agree with the Reviewer on the fact that a similar learning criterion does not rule out the possibility that attentional demands may vary between the two conditions, and that an absence of performance difference in one age group does not allow to infer about task complexity in the other age groups. As a consequence, we have removed this over-interpretation (as well as the associated figure) from the revised version of the manuscript.

7. Line 410: What was the rationale for adopting different visual acuity inclusion criteria for young (7/10) and older (5/10) adults?

Our study used participants from the Silversight cohort, which was created in 2015 by our laboratory Aging in Vision and Action at the Institute of Vision, in collaboration with the Clinical Investigation Center at the Quinze-Vingts National Ophthalmological Hospital, Paris. The SilverSight cohort counts <inline-graphic mimetype="image" mime-subtype="png" xlink:href="media/image1.png" />350 participants older than 18 years of age and without any pathology or deficit that could interfere with the visual, cognitive, hearing and vestibular functions. The entire cohort population underwent (and follow-ups are regularly done) an ophthalmological screening (conducted by medical doctors at CIC), a functional visual screening (conducted by orthoptists), an otorhinolaryngological examination (conducted by ENT specialists), and a static/dynamic balance assessment (conducted by podiatrists and posture specialists). Visual acuity naturally decreases with advancing age, even in the absence of pathology so the 5/10 criterion for visual acuity reflects this physiological process. All subjects were corrected to normal when performing our experiment (not necessarily with the best correction possible, but with the correction they usually wear when walking in an outdoor environment).

8. Why were neuropsychological, visual acuity, and post-task questionnaires only collected in a subset of participants?

For cognitive and visual measures, please our response in point 5 above.

Concerning the post-task questionnaire, we realized that getting the participant internal representation of space (by asking them to draw a map of the environment) could be interesting for our study after debriefing with the first old subjects enrolled. One of them, for instance, mentioned his/her (wrong) impression that landmarks were changed during the learning phase. We came up with a short questionnaire to evaluate this point, but we could only get a subset of participants. We started the enrollment of young adults before older adults and children, which explains why we have few post-task questionnaires in young adults. We think that post-task questionnaire data (small sample) and navigational/gaze/cognitive data (larger sample) all point in the direction of a landmark-specific deficit in older adults.

Some other issues the editor (Ekstrom) noted:1) "Egocentric strategies rely on spatial codes anchored on the subject's body, whereas allocentric strategies are grounded on representations that are independent from the subject's position and orientation, akin to a topographic map (4)."It should be noted that egocentric processing also can involve (and often does involve) visual snapshots of the environment, see for example (Waller and Hodgson 2006).

We thank the Editor. We have added this reference, which perfectly fits with some results of the paper.

2) "The study of age-related changes in human navigation has added a new temporal dimension to this research domain by investigating the evolution of spatial learning and wayfinding behavior across the lifespan."As this is a cross-sectional study (old vs. young vs. children) and not longitudinal, it seems difficult to rule out cohort effects. Perhaps instead phrase as age-related differences.

We agree that the use of terminology like “evolution” can be misleading. We have corrected every instance in the text (title/abstract/introduction). In the Discussion section, we have mentioned the need for longitudinal confirmation of the results observed here (lines 430-433).

3) "Hence, an alternative explanation consistent with the literature is that the widely-accepted hypothesis of age-related allocentric deficit may in fact reflect a landmark-processing impairment."It is unclear if what is being looked at is a "deficit" or a difference in processing/strategy.

Map drawing analyses, and to some extent gaze analyses, showed that landmark locations were incorrectly encoded by older adults. This, we believe, might be one of the reasons older subjects choose a different strategy during the probe trials.

We used the words ‘deficit’ and ‘impairment’ to describe older adults’ behavioral difficulties, as compared to young adults, in line with existing aging literature. We can indeed agree that older adults’ egocentric preference can be detrimental in real life situations, where they must be able to take detours and/or plan complex navigational paths.

When referring to our own results, we have preferred the term ‘strategy preference’ or ‘bias towards egocentric strategies’.

4) "We sought to test these hypotheses by employing a radial-maze experimental paradigm, traditionally used to dissociate egocentric and allocentric navigation in rodents (27,28) and humans (15)."From the way things are written, it is difficult to tell which findings came from the real-world maze and the virtual maze. This should be made clearer upfront and at all points in the results, this distinction should be clearer.

We understand the confusion, and we have clarified upfront how we used the data from each experiment. Please, see line 136 in the revised manuscript.

5) "We sought to test these hypotheses by employing a radial-maze experimental paradigm, traditionally used to dissociate egocentric and allocentric navigation in rodents (27,28) and humans (15).The sample size varies somewhat throughout the manuscript. It should be made clear throughout exactly what the N is in each comparison.

We have tried to specify the n in each analysis and to explain in the text why n varies. See for instance lines 131, 136, and 198 in the revised manuscript.

6) "Older adults required a higher number of trials than young adults to reach the learning criterion of 4 consecutive successful trials (Figure 3A; older vs. young adults: U=287.5, p<0.05).It is unclear what test is being conducted here and what the degrees of freedom are.

Two samples non-parametric comparison were done with the Mann-Whitney U test. The U test has no degree of freedom but we have additionally provided an effect size estimation (r), the sample size, and non-parametric Bayes factor for a better read out of these statistical tests.

7) "A subset of the participants in the virtual experiment underwent a complete battery of visual and neurocognitive screenings, resulting in 19 measurements per subjectHow many subjects were tested here?

Please, see the response to Reviewer 2 point 5 on this matter. We could get cognitive data from 64 participants, although only 47 of them had the complete 19 measurements used for the principal component analysis (PCA). With more subjects, our interpretation from the previous version are maintained.